# GRASP: Deterministic Argument Ranking
# in Natural-Language Debates via Attack–Defense Propagation

## Abstract

Large language models are increasingly deployed as automated judges to evaluate the strength of arguments. As this use increases, their legitimacy hinges on their accuracy, consistency, and transparency. However, we demonstrate that holistic judging—a commonplace practice where an LLM provides a global verdict on a debate—suffers from significant inter-model disagreement. We argue that this instability arises from collapsing a debate's complex interaction structure into a single score. To address this, we propose GRASP (Gradual Ranking with Attacks and Support Propagation), a deterministic framework that aggregates stable, local relational judgments, such as attacks and defenses, into a global ranking. We demonstrate that while holistic judgments are inconsistent, local interactions are significantly more reproducible across models, allowing GRASP to produce more stable rankings across diverse LLM backends. We further show that GRASP scores do not correlate with human "convincingness" labels. We highlight this as a vital sociotechnical distinction: while humans and holistic LLM judges are susceptible to rhetorical style, GRASP measures structural sufficiency—a defense-aware notion of argument robustness that evaluates how well a position is defended against explicit rebuttals. Overall, we propose GRASP to provide a more principled, transparent, and auditable alternative for automated debate judging.

## 1. Introduction

Large language models (LLMs) are increasingly used not only to generate content, but also to act as automated judges for evaluating discourse quality, moderating debates (Chuang et al., 2025), and supporting multi-agent decision-making (Bai et al., 2022; Ouyang et al., 2022; Achiam et al., 2023; Liang et al., 2022). For such systems to be credible arbiters of deliberation, their evaluations must be consistent, transparent, and grounded in argumentative merit rather than model-specific idiosyncrasies.

Most current practice relies on *holistic judging*, where an LLM is presented with an entire debate and asked to output a global verdict or ranking (Zheng et al., 2023; Liang et al., 2022; Thakur et al., 2025). We show that these global judgments exhibit substantial inter-model disagreement, consistent with growing concerns about the reliability of LLM-as-a-judge paradigms (Ye et al., 2024; Chehbouni et al., 2025). Such instability suggests that holistic judging is driven in part by rhetorical and stylistic preferences—e.g., biases, verbosity, tone, or narrative coherence (Taubenfeld et al., 2024; Stureborg et al., 2024; Hu et al., 2024)—rather than shared normative principles.

We argue that this reflects a structural limitation of the paradigm: global judgments collapse rich dialectical interactions into a single black-box score. Instead, we ground evaluation in *local semantic interactions*, drawing on computational argumentation and abstract argumentation frameworks (Dung, 1995; Besnard & Hunter, 2001; Bench-Capon et al., 2007; Wachsmuth et al., 2017). While models may disagree on holistic verdicts, they are markedly more consistent on local pairwise judgments (Nie et al., 2020; Bowman et al., 2015; Liu et al., 2024), making such interactions a more reliable primitive.

Motivated by this observation, we introduce GRASP (**G**radual **R**anking with **A**ttacks and **S**upport **P**ropagation), a convergent propagation algorithm that composes local attack and support judgments into a global ranking by aggregating direct attacks and higher-order defenses on an explicit interaction graph.

We further introduce *structural sufficiency*, a defense-aware notion of argument robustness defined relative to an explicit attack–support graph. Structural sufficiency is closest to global sufficiency (Cohen, 2001; Gurcke et al., 2021), but evaluates robustness only with respect to the instantiated structure, rather than all counterarguments that could be

[1]Anonymous Institution, Anonymous City, Anonymous Region, Anonymous Country. Correspondence to: Anonymous Author <anon.email@domain.com>.

Preliminary work. Under review by the International Conference on Machine Learning (ICML). Do not distribute.

anticipated. It is related to gradual and ranking-based argumentation semantics (Baroni et al., 2011; Amgoud & Ben-Naim, 2013).

Empirically, we show that GRASP rankings are substantially more reproducible across models than direct LLM judging, yet largely uncorrelated with human "convincingness" labels, indicating that GRASP captures a notion of robustness complementary to persuasive effectiveness.

**Contributions.**

- We document substantial inter-model disagreement for holistic LLM-based judging.

- We introduce **GRASP**, a convergent propagation algorithm for structural argument ranking.

- We formalize **structural sufficiency**, a defense-aware notion of argument robustness over explicit interaction graphs.

- We show that GRASP-based rankings are highly reproducible across models while capturing a notion of robustness distinct from convincingness.

## 2. GRASP: A Structural Strength Propagation Operator

Motivated by the unreliability of holistic argument evaluation and the relative consistency of local relational judgments, we introduce GRASP (Gradual Ranking with Attacks and Support Propagation), an iterative operator that computes continuous argument strengths from an explicit interaction graph. GRASP aggregates local attack and defense relations into a global ranking while remaining agnostic to rhetorical appeal or hypothetical objections.

### 2.1. Background: Abstract Argumentation and Ranking Semantics

We build on *Abstract Argumentation Frameworks* (AAFs), where arguments are nodes and attacks are directed edges. Classical AAF semantics focus on set-valued notions of acceptability, which are ill-suited for fine-grained comparison.

Ranking-based semantics address this by assigning each argument a numerical strength. A prominent example is the *H-categorizer* (Besnard & Hunter, 2001), which penalizes arguments according to the total strength of their attackers. While effective as a local heuristic, such methods treat attacks independently and do not model defensive structure, i.e., how arguments protect one another by countering shared attackers.

Our aim is to retain the simplicity and interpretability of

attack-based rankings while incorporating defense in a principled and scalable way.

### 2.2. Weighted Attack and Defense Structure

Let $A = \{a_1, \ldots, a_n\}$ be a set of arguments. We represent the interaction structure using two weighted matrices operating on argument strength vectors $\mathbf{s}^{(t)} \in \mathbb{R}^n_{\geq 0}$:

- $W \in [0, 1]^{n \times n}$, the *attack matrix*, where $W_{ij}$ quantifies the strength with which argument $a_i$ attacks $a_j$.

- $D \in [0, \infty)^{n \times n}$, the *defense matrix*, where $D_{kj}$ quantifies the extent to which argument $a_k$ contributes to the defense of $a_j$.

### 2.3. The GRASP Update Rule

Given an initial strength vector $\mathbf{s}^{(0)} \in \mathbb{R}^n_{\geq 0}$, GRASP proceeds by iteratively updating argument strengths through a nonlinear operator that balances the weakening effect of attacks against the reinforcing effect of defense.

**Undamped operator.** We first define the (undamped) GRASP operator $G : \mathbb{R}^n \to \mathbb{R}^n$ coordinatewise as

$$G(s)_j \;=\; \frac{1 + \beta \sum_k D_{kj} s_k}{1 + \alpha \sum_i W_{ij} s_i}, \qquad (1)$$

where the denominator penalizes argument $a_j$ according to the total strength of its attackers, while the numerator rewards it based on the strength of its defenders. The parameters $\alpha, \beta \geq 0$ control the relative influence of attack and defense.

---

**GRASP Update Rule**

Define the damped GRASP operator $\widehat{G} : \mathbb{R}^n \to \mathbb{R}^n$ as

$$\widehat{G}(s) \;=\; (1 - \gamma)s + \gamma\, G(s), \qquad \gamma \in (0, 1]. \quad (2)$$

The GRASP iteration is then given by

$$s^{(t)} \;=\; \widehat{G}\big(s^{(t-1)}\big), \qquad (3)$$

or equivalently, coordinatewise,

$$s_j^{(t)} = (1 - \gamma)s_j^{(t-1)} + \gamma \cdot \frac{1 + \beta \sum_k D_{kj} s_k^{(t-1)}}{1 + \alpha \sum_i W_{ij} s_i^{(t-1)}}. \qquad (4)$$

---

**Damped GRASP operator and iteration.** To improve numerical stability and guarantee convergence in dense or highly cyclic interaction structures, we employ a damped

version of the operator, analogous to relaxation schemes in iterative optimization.

Repeated application of this update yields a stable strength assignment that reflects how well each argument withstands attack within the explicit interaction structure. Final rankings are obtained by ordering arguments according to their converged strength values.

### 2.4. Interpretation

GRASP generalizes attack-based ranking methods such as the H-categorizer by explicitly incorporating defense while preserving locality and interpretability. The resulting scores operationalize the notion of *structural sufficiency* introduced in Section 4: arguments are strong to the extent that their attackers are neutralized by available counter-attacks in the instantiated structure.

The GRASP operator itself is agnostic to how defensive influence is computed. Specific choices of the defense matrix $D$ reflect modeling and computational trade-offs and are discussed in the experimental section.

## 3. Convergence of the GRASP Operator

We now analyze the dynamics induced by the GRASP update rule introduced in 2.3. The GRASP iteration defines a nonlinear operator on argument strength vectors, and understanding its stability, fixed points, and convergence behavior is essential for interpreting the structural rankings it produces.

In Appendix B, we study the GRASP update as a map $G : \mathbb{R}^n \to \mathbb{R}^n$, induced by the non-symmetric weighted attack matrix $W$ and the derived defense matrix $D$. The nonlinearity of the operator arises from the interaction between attack aggregation in the denominator and defense propagation in the numerator, reflecting the dialectical coupling between opposition and reinstatement.

**Theorem 3.1.** *Let $\mathcal{S} := \{s \in \mathbb{R}^d, \|s - 1\|_\infty \leq 1\}$ and let $G : \mathcal{S} \to \mathbb{R}^d$ be defined elementwise by $G(s)_i = \frac{1 + \beta(D^\top s)_i}{1 + \alpha(W^\top s)_i}$. If $W, D$ have non-negative entries and*

$$\alpha \leq \frac{1}{4\|W\|_1}, \qquad \beta \leq \frac{1}{4\|D\|_1},$$

*with $\|A\|_1 := \max_j \sum_i |a_{ij}|$, then $G(\mathcal{S}) \subseteq \mathcal{S}$ and $G$ is a contraction on $\mathcal{S}$. Consequently, $G$ admits a unique fixed point $s^* \in \mathcal{S}$, and the iteration $s_{k+1} = G(s_k)$ converges to $s^*$. The result extends to the damped variant in Eq. 2.*

Our proof in Appendix B uses standard tools but applies them to a non-standard and non-linear setup. Given the worst-case nature of our analysis, we note that the coefficients $\alpha$ and $\beta$ suggested by the proof might not yield the best-performing results in general, and hence treat them as tuning parameters in our experiments (see e.g. Section 5.3).

## 4. Structural Sufficiency

The GRASP operator introduced in the previous section computes a stable equilibrium of argument strengths based solely on explicit attack and support relations. We now clarify *what notion of argument quality this equilibrium corresponds to*. Within the argumentation literature, the closest conceptual analogue is *global sufficiency* (Gurcke et al., 2021; Cohen, 2001): the idea that an argument is strong if it adequately withstands opposing arguments. However, global sufficiency is typically defined relative to objections that *could be anticipated*, including hypothetical or implicit counterarguments. In contrast, GRASP operates strictly on the instantiated interaction structure.

This motivates introducing the following variant:

**Structural sufficiency (in plain English).** We define *structural sufficiency* as a notion of argument robustness that depends only on the *explicit* attack–support structure present in a debate, abstracting away from rhetorical appeal, persuasion, or imagined objections.

We proceed with a few definitions formalizing this notion.

**Structure.** An *argumentation structure* is a tuple $\mathcal{G} = (A, R^-, R^+)$, where $A$ is a finite set of arguments, $R^- \subseteq A \times A$ is an attack relation, and $R^+ \subseteq A \times A$ is a support relation. We write $(b, a) \in R^-$ as "$b$ attacks $a$" and $(c, a) \in R^+$ as "$c$ supports $a$."

**Neutralization.** Fix an argument $a \in A$ and an attacker $b \in A$ such that $(b, a) \in R^-$. We say that the attack $(b, a)$ is *structurally neutralized* in $\mathcal{G}$ if there exists at least one argument $c \in A$ such that $(c, b) \in R^-$. That is, an attack is neutralized whenever the attacker itself is explicitly attacked within the structure.

Support relations may enable potential defenders (e.g., arguments that support $a$ may serve as available attackers of $b$), but structural sufficiency does not require any specific operational treatment of support edges.

**Structural sufficiency.** An argument $a$ is *structurally sufficient* in $\mathcal{G}$ if every explicit attack on $a$ is neutralized:

$$\text{SS}(a; \mathcal{G}) \iff \forall b \in A, \ (b, a) \in R^-$$
$$\Rightarrow \exists c \in A \text{ such that } (c, b) \in R^-.$$

All quantification ranges only over arguments explicitly present in $A$. Thus, unlike *global sufficiency*—which concerns rebutting counterarguments that *could be anticipated*—structural sufficiency evaluates robustness strictly with respect to the instantiated interaction structure (Wachsmuth et al., 2017).

**Axioms of Structural Sufficiency.** Our definitions above allow us to specify the following criteria, which we dub *axioms*, that formalize the minimal requirements for any interaction-based robustness criterion based on structural sufficiency.

**Axiom S1 (Attack Sensitivity).** Unneutralized attacks invalidate sufficiency:

$$(b, a) \in R^- \ \wedge \ \neg \exists c \in A : (c, b) \in R^- \ \Rightarrow \ \neg \mathrm{SS}(a; \mathcal{G}).$$

**Axiom S2 (Defense Reinstatement).** Attacking an attacker restores sufficiency with respect to that attack:

$$(c, b) \in R^- \ \wedge \ (b, a) \in R^- \ \Rightarrow \ (b, a) \text{ is neutralized.}$$

**Axiom S3 (Structural Locality).** Only structurally connected arguments affect sufficiency. If no directed path exists from $x$ to $a$ in $(A, R^- \cup R^+)$, then

$$x \text{ has no effect on } \mathrm{SS}(a; \mathcal{G}).$$

**Axiom S4 (Baseline Sufficiency).** Arguments without attackers are sufficient by default:

$$\neg \exists b \in A \text{ s.t. } (b, a) \in R^- \ \Rightarrow \ \mathrm{SS}(a; \mathcal{G}).$$

Together, these axioms define a notion of robustness that depends only on the explicit interaction structure. Structural sufficiency is Boolean: an argument is sufficient iff all its explicit attackers are countered in the graph. While conceptually simple, this discrete criterion does not support graded comparison. GRASP aggregates the same primitives—explicit attacks, counter-attacks as defense, and structural locality—but replaces binary neutralization with smooth propagation and normalization. In this sense, GRASP can be viewed as a *continuous relaxation* of structural sufficiency that yields graded strength scores.

# 5. Experiments

## 5.1. StructDebate

We introduce STRUCTDEBATE, a controlled synthetic debate dataset designed to study *structural evaluation* and argument ranking under explicitly instantiated interaction regimes. The dataset is constructed to isolate the effects of attack–defense structure from rhetorical style, enabling reproducible analysis of structural ranking methods.

STRUCTDEBATE consists of machine-generated arguments grounded in real-world debate motions, with controlled prompts, roles, and semantic constraints. Unlike existing argument quality datasets, the focus is not on human judgments or persuasive effectiveness, but on recoverable structural relations between arguments.

### 5.1.1. DEBATE MOTIONS

We sample 50 motions from the public DebateData.io corpus.[1] Motions span public policy, economics, technology, law, and ethics, and are phrased as binary propositions suitable for adversarial debate. Each motion defines an independent debate instance.

### 5.1.2. ARGUMENT GENERATORS

Arguments are generated using a diverse set of large language models: `openai/gpt-5.2-pro`, `anthropic/claude-opus-4.5`, `mistralai/mistral-small-creative`, `qwen/qwen3-max`, and `x-ai/grok-4`. These models are used exclusively as *generators* and never as judges unless explicitly stated. Prompts are standardized across models to minimize stylistic variance.

### 5.1.3. DEBATE SETTINGS

We generate arguments under two complementary settings.

**Pool Setting.** Arguments are generated independently. For each motion, side (PRO or CON), and semantic angle, models produce short standalone arguments, yielding an unordered pool that isolates intrinsic content from interaction effects.

**Multi-Turn Setting (Self-Debate).** Arguments are generated sequentially in a 10-turn structured self-debate, with turns alternating between PRO and CON. The same model generates both sides. Each turn observes the full debate history, inducing explicit attack and defense relations while avoiding multi-agent confounds.

### 5.1.4. SEMANTIC ANGLES

Each argument is generated with respect to one of six semantic angles: ECONOMIC, LEGAL, MORAL, POLITICAL, SOCIAL, and TECHNOLOGICAL. Angles are balanced across sides and rotated across turns in the multi-turn setting.

## 5.2. Dataset Statistics

STRUCTDEBATE contains **7,000 arguments** across **300 debates**: **50 pool debates** (40 arguments each) and **250 multi-turn debates** (20 arguments each). The dataset is balanced by stance and angle.

## 5.3. Inter-Model Agreement via Structural Aggregation

A central motivation for structural evaluation is that different judge models often produce highly inconsistent argument rankings when operating directly on raw text. We

---

[1] https://debatedata.io/

|  | Pool | Multi-turn |
|---|---|---|
| # Debates | 50 | 250 |
| # Arguments | 2,000 | 5,000 |
| Arguments / debate (mean) | 40.0 | 20.0 |
| Mean length (tokens) | 42.5 | 94.9 |
| Turns per debate | – | 10 |

*Table 1.* Summary statistics for STRUCTDEBATE.

test whether explicitly aggregating pairwise attack relations into a global interaction structure can induce more stable rankings across models.

**Constructing the attack graph.** For each debate, we instantiate a fully connected directed graph over arguments. Inspired by natural language inference (NLI), each ordered pair $(a_i, a_j)$ is scored for contradiction using an LLM-based NLI prompt that asks by what strength does argument $a_i$ contradicts or attacks argument $a_j$. The resulting contradiction probability is treated as a weighted attack from $i$ to $j$, yielding a dense attack matrix $W$.

Each judge model independently produces its own $W$ using the same prompting template. GRASP then converts each $W$ into a global ranking by computing argument strengths from the induced structure and sorting arguments by their final scores.

**Baselines and judge models.** We compare against RAW ranking, where a judge model directly produces a global ranking from the full debate. We use six judge models:

- `anthropic/claude-haiku-4.5`
- `deepseek/deepseek-v3.2`
- `google/gemini-3-flash-preview`
- `meta-llama/llama-4-scout`
- `openai/gpt-5.2-chat`
- `xiaomi/mimo-v2-flash`

In GRASP, these models are used only to score local attack relations, which are then structurally aggregated.

**GRASP variants.** We evaluate five GRASP variants that differ only in how the attack matrix $W$ is globally normalized and whether the defense matrix $D$ is rescaled. All variants use the same GRASP update rule; only the preprocessing of $W$ and $D$ changes.

We define the defense matrix as $D = WW$ to capture two-hop counter-attacks, i.e., an argument defends another by attacking its attacker, which is the minimal higher-order interaction required for modeling defense. All runs initialize strengths uniformly with $s^{(0)} = \mathbf{1}$. Unless otherwise stated, hyperparameters are fixed *a priori* to $\alpha = 1.0$, $\beta = 0.6$, and damping $\gamma = 0.9$; a post-hoc sensitivity analysis over $(\alpha, \beta, \gamma)$ is reported in Appendix G.

| Method | $\tau$ | $\Delta\tau$ | Swap | $\rho$ | Top-3 | Top-5 |
|---|---|---|---|---|---|---|
| **Multi-turn Setting** | | | | | | |
| RAW | 0.309 | – | 0.345 | 0.380 | 0.410 | 0.474 |
| GRASP | 0.626 | +0.317 | 0.187 | 0.779 | 0.487 | 0.619 |
| GRASP-W$_\infty$ | 0.626 | +0.317 | 0.187 | 0.779 | 0.486 | 0.620 |
| GRASP-W$_1$ | 0.607 | +0.298 | 0.197 | 0.768 | 0.525 | 0.634 |
| GRASP-W$_\infty$+$\bar{D}$ | 0.593 | +0.284 | 0.204 | 0.755 | 0.528 | 0.632 |
| GRASP-W$_1$+$\bar{D}$ | 0.575 | +0.266 | 0.212 | 0.740 | 0.542 | 0.635 |
| **Pool Setting** | | | | | | |
| RAW | 0.337 | – | 0.331 | 0.425 | 0.385 | 0.417 |
| GRASP | 0.623 | +0.286 | 0.189 | 0.780 | 0.509 | 0.574 |
| GRASP-W$_\infty$ | 0.623 | +0.286 | 0.188 | 0.781 | 0.509 | 0.574 |
| GRASP-W$_1$ | 0.604 | +0.267 | 0.198 | 0.772 | 0.528 | 0.587 |
| GRASP-W$_\infty$+$\bar{D}$ | 0.591 | +0.254 | 0.205 | 0.761 | 0.530 | 0.584 |
| GRASP-W$_1$+$\bar{D}$ | 0.580 | +0.243 | 0.210 | 0.752 | 0.535 | 0.593 |

*Table 2.* Setting-wise inter-model agreement (averaged per debate). $\Delta\tau$ denotes absolute improvement over RAW within each setting.

- **GRASP**: No normalization. Uses $W$ as produced by the LLM and $D = WW$.

- **GRASP-W$_\infty$**: Global $L_\infty$ normalization of $W$ before computing $D$.

- **GRASP-W$_1$**: Global $L_1$ normalization of $W$ before computing $D$.

- **GRASP-W$_\infty$+$\bar{D}$**: Global $L_\infty$ normalization of $W$, followed by computation of $D = WW$ and rescaling of $D$ to obtain $\bar{D}$.

- **GRASP-W$_1$+$\bar{D}$**: Global $L_1$ normalization of $W$, followed by computation of $D = WW$ and rescaling of $D$ to obtain $\bar{D}$.

In all cases, GRASP is run with the same update rule using the pair $(W, D)$ (or $(W, \bar{D})$ for the rescaled variants). Thus, any differences in behavior are attributable solely to the scale of interaction weights rather than to changes in the ranking dynamics.

**Agreement metrics.** Within each debate, we compute pairwise agreement between judge rankings using Kendall's $\tau$, Spearman's $\rho$, normalized Kendall swap distance, and Top-$k$ overlap, and average the resulting scores across debates.

**Results.** Table 2 shows that GRASP-based structural aggregation more than doubles inter-model agreement relative to RAW rankings in both pool and multi-turn settings. The base GRASP and GRASP-W$_\infty$ variants achieve the strongest and most stable performance ($\tau \approx 0.62$), indicating that simple global normalization of attack strengths is sufficient.

| Setting | Method | Borda dist. | Kemeny dist. | Outlier Model |
|---|---|---|---|---|
| Multi-turn | GRASP | 27.2 | 27.3 | Llama-4 Scout |
| | GRASP-$W_\infty$ | 27.2 | 27.3 | Llama-4 Scout |
| | GRASP-$W_1$ | 28.6 | 28.8 | Llama-4 Scout |
| | GRASP-$W_\infty+\bar{D}$ | 29.4 | 29.5 | Llama-4 Scout |
| | GRASP-$W_1+\bar{D}$ | 30.8 | 30.9 | Llama-4 Scout |
| | RAW | 48.7 | 49.2 | GPT-5.2 Chat |
| Pool | GRASP | 116.3 | 116.5 | Llama-4 Scout |
| | GRASP-$W_\infty$ | 116.4 | 116.6 | Llama-4 Scout |
| | GRASP-$W_1$ | 123.8 | 124.0 | Llama-4 Scout |
| | GRASP-$W_\infty+\bar{D}$ | 129.1 | 128.9 | Llama-4 Scout |
| | GRASP-$W_1+\bar{D}$ | 133.0 | 132.9 | Llama-4 Scout |
| | RAW | 204.2 | 204.8 | GPT-5.2 Chat |

*Table 3.* Farthest-from-consensus model (largest mean Kendall distance to consensus). Lower is better. Structural aggregation via GRASP strongly reduces worst-case divergence relative to RAW.

### 5.4. Consensus Divergence

Beyond pairwise agreement, we study how far individual judge models deviate from a global consensus ranking. For each debate and aggregation method, we form two standard consensus rankings across judges: (i) *Borda* aggregation and (ii) a greedy approximation to the *Kemeny*-optimal ranking. For each judge model, we compute its Kendall swap distance to the corresponding consensus ranking. We then report, for each setting and method, the judge model with the largest mean distance to consensus (the *farthest-from-consensus* model).

All GRASP variants use the same underlying *strength-based structural scoring*: each argument receives a score based on the total weighted strength of incoming attacks and the total weighted strength of incoming defenses, derived from the attack matrix $W$ and its induced defense matrix $D = WW$. Different GRASP variants differ only in whether and how $W$ (and optionally $D$) are normalized prior to scoring.

**Results.** Table 3 reports the mean distance of the farthest-from-consensus judge for each method and setting. Across both multi-turn and pool debates, all GRASP variants substantially reduce worst-case divergence compared to RAW rankings. In particular, the most divergent GRASP judge is consistently around $2\times$ closer to consensus than the most divergent RAW judge.

Notably, this reduction in worst-case ranking divergence aligns with the high pairwise similarity of the underlying attack graphs across NLI judges (Section 5.5), suggesting that agreement at the level of local pairwise judgments translates into increased robustness of global structural aggregation.

**Takeaway.** GRASP improves not only average agreement but also *robustness*: even the most divergent judge under GRASP remains substantially closer to consensus than the most divergent RAW judge.

### 5.5. Geometry of Attack Graphs and GRASP Dynamics

A central premise of GRASP is that it operates over a stable interaction geometry between arguments rather than relying

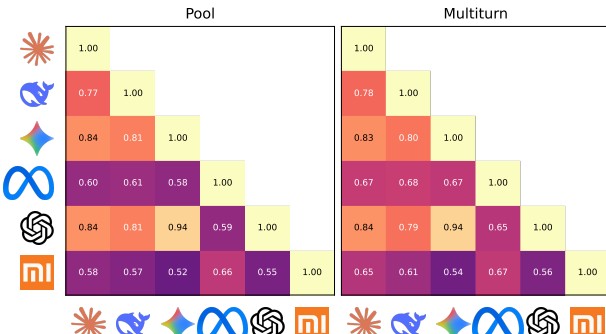

*Figure 1.* Pairwise mean Pearson correlation between attack-weight matrices $W$ induced by different NLI judge models. Left: pool setting. Right: multi-turn setting. Each cell reports the average Pearson correlation between vectorized off-diagonal entries of $W$ across debates, measuring agreement in relative attack patterns.

on idiosyncrasies of any single judge model. We test this by analyzing both the similarity of induced attack graphs across judges and the resulting convergence behavior of GRASP.

Figure 1 shows that different NLI judges induce strongly correlated attack-weight matrices $W$, with most pairwise Pearson correlations lying in the range 0.55–0.95 across both pool and multi-turn settings. The qualitative structure of the similarity matrix is consistent across settings, indicating that judges largely agree on the *relative pattern* of which argument pairs are adversarial, even if they differ in absolute edge magnitudes.

Figure 2 relates this geometry to GRASP dynamics. While convergence does not follow a simple function of graph density, models occupy distinct and persistent regions of the density–iteration plane. These relative placements are preserved between GRASP and GRASP-$W_\infty$, showing that global normalization of $W$ does not materially change the underlying interaction regime.

Together, these results indicate that GRASP operates on a largely model-agnostic interaction structure, which explains the high cross-model agreement of its rankings.

**Takeaway.** Different NLI judges induce geometrically similar attack graphs, and this shared geometry governs GRASP's convergence behavior. GRASP's consistency therefore arises from properties of the induced graphs rather than from properties of any particular judge.

### 5.6. Case Study: Graph Structure and Rank Dynamics in a Single Debate

We present a qualitative case study on the multi-turn debate `mt_048_x_ai_grok-4` with motion: *"This House would break up dominant technology monopolies."* The goal is to illustrate (i) how different NLI judges induce different *geometries under high-confidence attack thresholds*, and

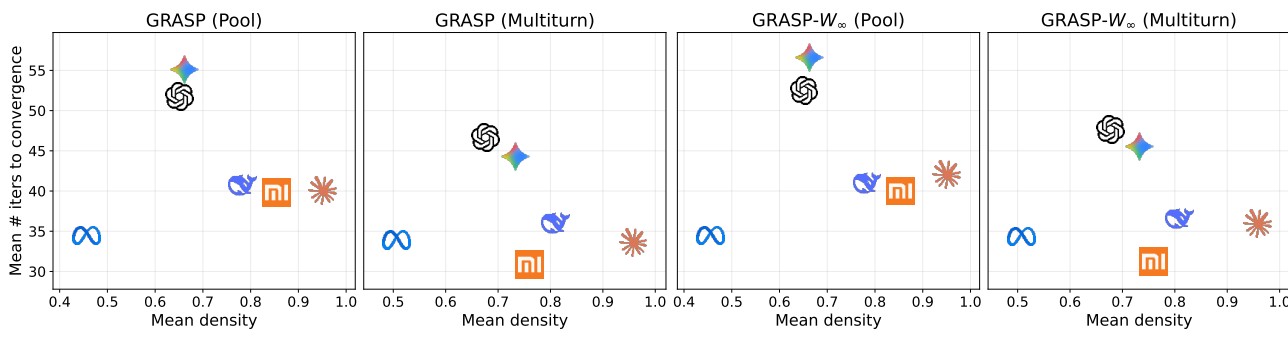

*Figure 2.* Convergence vs. attack-graph density for GRASP and GRASP-$W_\infty$. Each logo denotes one NLI judge model, averaged across debates.

(ii) how GRASP's iterative updates translate such structure into non-trivial rank dynamics. The texts of the arguments referenced in this section are provided in Appendix I.

**Attack-graph structure across NLI judges.** Figure 3 visualizes thresholded attack graphs ($W_{ij} > \tau$, $\tau = 0.6$) for the same debate under six NLI judges. While all graphs are constructed from identical argument sets, their high-threshold structure differs in both connectivity and organization.

Some judges yield highly dense high-confidence graphs (absolute off-diagonal density $d \approx 0.95$–$0.98$), whereas others produce noticeably sparser graphs ($d \approx 0.5$–$0.65$). Moreover, the mean positive attack strength $\mu$ varies substantially, implying that judges differ not only in how many attacks they predict, but also in how strong those attacks tend to be.

These differences emerge *after thresholding*, suggesting that disagreement is concentrated in which attacks are considered sufficiently strong, rather than in the existence of weak, noisy relations. This aligns with the view that local pairwise judgments are relatively reliable, but that global graph geometry is sensitive to how aggressively a judge assigns high-confidence contradictions.

**Rank dynamics under GRASP.** Using the attack graph produced by `openai/gpt-5.2-chat`, we track GRASP scores over iterations and visualize the rank trajectories of the most volatile arguments (Figure 4, $\tau = 0.5$ for graph construction). Several arguments undergo large early rank shifts (e.g., changes of $\Delta$rank $\approx 10$–$18$) before stabilizing.

The trajectories are not monotonic: some arguments temporarily rise before being demoted, while others steadily improve. This reflects the coupled nature of the GRASP update, where an argument's score depends on both incoming attacks and higher-order interactions propagated through the graph. Convergence therefore arises from coordinated global reweighting rather than from a simple local sorting heuristic.

**Takeaway.** Even within a single debate, high-confidence attack structure varies across NLI judges, leading to dif-

ferent graph geometries on which GRASP operates. These geometries, in turn, induce non-trivial and interpretable rank dynamics. The case study supports the view that GRASP is sensitive to genuine structural signal encoded by local pairwise judgments, rather than merely smoothing raw rankings.

## 5.7. Case Study: Structural Consensus vs. RAW Disagreement

We present a representative case in which an argument is ranked near-unanimously among the top positions by GRASP across different judge models, yet receives highly dispersed and often low rankings under direct RAW judging. This contrast illustrates how aggregating local pairwise interactions into an explicit structure stabilizes global priorities, even when individual judges disagree substantially at the level of holistic evaluation.

**Topic.** This House would require warrants for searches instead of allowing stop-and-frisk.

---

**GRASP (Structural Consensus)**

**Generator:** openai/gpt-5.2-pro
**Stance:** Pro
**Angle:** Political
**Turn:** 4
**# Attackers:** 10    **Mean attack strength:** 0.422

**Argument:** *Politically, requiring warrants shifts the authority to search from unilateral street-level discretion to a process that includes independent oversight, reinforcing separation of powers and democratic control over coercive state action. This reduces the risk that search practices become informal policy tools shaped by electoral pressures or internal quotas rather than publicly accountable standards. Clear warrant rules also create more consistent statewide governance, limiting local variations that can undermine legitimacy and deepen political polarization over policing.*

---

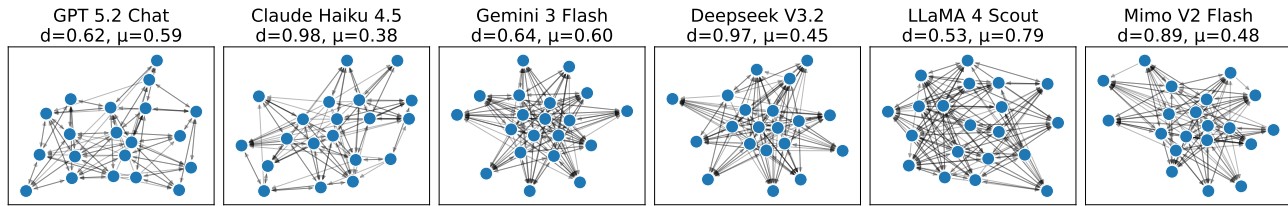

*Figure 3.* Attack graphs for the same debate under different NLI judges ($W_{ij} > \tau$, $\tau = 0.6$). Each subplot reports absolute density $d = \frac{1}{n(n-1)} \sum_{i \neq j} \mathbb{I}[W_{ij} > 0]$, and mean off-diagonal attack strength $\mu = \frac{1}{n(n-1)} \sum_{i \neq j} W_{ij}$.

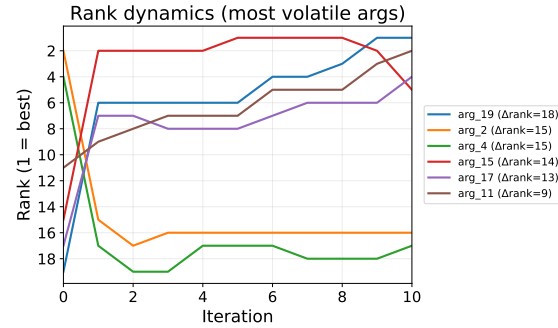

*Figure 4.* Rank dynamics of the most volatile arguments for `mt_048_x-ai_grok-4` under GRASP (NLI judge: `openai/gpt-5.2-chat`).

> ## GRASP Rankings
>
> anthropic/claude-haiku-4.5: 1
> deepseek/deepseek-v3.2: 1
> google/gemini-3-flash-preview: 1
> meta-llama/llama-4-scout: 1
> openai/gpt-5.2-chat: 1
> xiaomi/mimo-v2-flash: 2

> ## RAW Rankings
>
> anthropic/claude-haiku-4.5: 7
> deepseek/deepseek-v3.2: 17
> meta-llama/llama-4-scout: 5
> openai/gpt-5.2-chat: 20
> xiaomi/mimo-v2-flash: 19

**Analysis.** This argument achieves **GRASP top-3 fraction = 1.00 (6/6)** but **RAW top-3 fraction = 0.00 (0/5)**, with RAW ranks exhibiting high dispersion. Although RAW judges emphasize surface-level salience or stylistic preferences, GRASP identifies this argument as structurally central because it attracts many attacks[2] from across angles yet remains defensible, signaling a pivotal dialectical position. This highlights how structural aggregation prioritizes glob-

---

[2]# Attackers and Mean attack strength are obtained from the $W$ constructed by the `openai/gpt-5.2-chat` judge.

ally consequential arguments rather than locally persuasive ones.

Additional ablations, prompt specifications, GRASP pseudocode, and analyses demonstrating the lack of correlation between GRASP scores and convincingness are provided in the Appendix.

## 6. Conclusion and Discussion

We introduced **GRASP**, a structural aggregation framework that ranks arguments using only their pairwise interaction patterns. Unlike quality-aware or persuasion-oriented models, GRASP treats *argument strength* as an emergent property of the attack graph: arguments gain strength solely through how they are attacked and defended. This makes GRASP a method for *structure-based evaluation*, rather than a predictor of rhetorical effectiveness. As a result, GRASP provides an explicit and auditable pathway from local interactions to global rankings.

Empirically, we show that GRASP substantially improves inter-model agreement, reduces worst-case divergence from consensus, and yields consistent rankings across independently constructed attack graphs. Furthermore, our results indicate that structural sufficiency and persuasive success are distinct properties, suggesting that logical robustness and rhetorical appeal capture complementary aspects of argument quality.

Compared to prior graph-based semantics and quality-aware frameworks (reviewed in Appendix A), GRASP uses no intrinsic node quality, no supervision, and no static centrality: all information arises from local pairwise interactions and their propagation.

**Limitations and Future Work.** Current limitations include $O(n^2)$ interaction cost, defense modeled only via two-hop counterattack ($D = WW$), and lack of human annotations for structural sufficiency. Future work includes scalable edge filtering, alternative defense formulations, human validation of structural sufficiency, using GRASP as a reward signal for training or steering multi-agent debate systems, and extending to streaming settings where arguments arrive over time and the interaction graph evolves.

## Impact Statement

This work advances understanding of how *structural sufficiency* can be modeled and evaluated independently of persuasion, rhetoric, or surface-level language features. By introducing GRASP, we provide a principled operator for deriving argument strength from explicit interaction structure alone, enabling transparent and interpretable analysis of multi-argument evaluation.

**Positive Impacts.** GRASP offers a foundation for building and auditing systems that reason over debates, legal arguments, policy discussions, and scientific claims in a structurally grounded manner. Because GRASP operates on explicit attack relations rather than latent heuristics, it facilitates inspection of why certain arguments are ranked highly, which may support applications in explainable AI, deliberative decision support, and tools for assisting human analysts in navigating complex argumentative corpora. The framework may also enable new evaluation methodologies for multi-agent debate and reasoning systems by decoupling structural coherence from persuasive success.

**Potential Risks and Misuse.** Structural ranking does not capture truth, ethical correctness, or societal desirability. Misinterpreting GRASP scores as indicators of factual validity or moral rightness could lead to inappropriate reliance in high-stakes contexts. Additionally, if attack graphs are constructed from biased or low-quality interaction scorers, downstream rankings may inherit these biases.

**Mitigations.** We emphasize that GRASP is intended as a *structural evaluator*, not a truth or persuasion oracle. It should be used in conjunction with complementary evaluation signals (e.g., factual verification, human judgment, or domain-specific constraints). All components of the pipeline are explicit and auditable, allowing practitioners to inspect and correct problematic edges or scoring behaviors.

Overall, we believe this work contributes positively by clarifying what can and cannot be inferred from argument structure, and by encouraging the development of analytical systems that are more transparent, modular, and theoretically grounded.

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

# Appendix

## A. Related Work

GRASP connects two research threads: (i) gradual and ranking-based semantics in abstract argumentation, and (ii) neural and large language model (LLM) approaches for inducing argumentative structure from text.

**Gradual Semantics and Argument Ranking.** Abstract argumentation frameworks (AFs) model arguments as nodes and attacks as edges, with classical semantics defined via accepted *extensions* (Dung, 1995). To enable fine-grained comparison, **gradual semantics** assign each argument a real-valued strength rather than a binary status. Foundational work introduced graduality into argumentation (Cayrol & Lagasquie-Schiex, 2005), followed by ranking and propagation-based semantics such as categorizer approaches (Bench-Capon, 2003; Baroni et al., 2011; Amgoud & Ben-Naim, 2013; Baroni et al., 2018). Subsequent research extends these ideas to weighted and quantitative settings (Libman et al., 2024), assumption-based argumentation (Rapberger et al., 2025), extension-derived rankings (Bengel et al., 2025), equilibrium-style numerical semantics (Gabbay & Rodrigues, 2015), and power-index-based formulations (Bistarelli & Taticchi, 2020). A complementary line studies axiomatic properties that ranking semantics should satisfy (Amgoud & Ben-Naim, 2013; Baroni et al., 2014).

GRASP contributes a damped iterative scoring operator with a contraction-based convergence guarantee, positioning it within numerical semantics while emphasizing large-scale empirical grounding.

**Grounding Argumentation with Language Models.** Argument mining aims to extract argumentative components and relations from text (Potash & Rumshisky, 2017; Chakrabarty et al., 2019; Wei et al., 2016; Mirzakhmedova et al., 2023), often targeting convincingness (Li et al., 2020; Potash et al., 2019; Bozdag et al., 2025) or argument quality prediction (Toledo et al., 2019; Wachsmuth & Werner, 2020; Wachsmuth et al., 2017). However, neural models can achieve high performance using shallow cues rather than genuine reasoning (Niven & Kao, 2019). Recent surveys document the rapid adoption of LLMs for argument-related tasks (Chen et al., 2024; Li et al., 2025; Mirzakhmedova et al., 2024; Rescala et al., 2024; Li et al., 2024). Several works describe argument structure as a feature for persuasion modeling (Stab & Gurevych, 2014), while others explore debate-style or multi-turn reasoning with LLMs (Zhang et al., 2025; Sanayei et al., 2025).

GRASP follows a hybrid strategy: neural models estimate *local* interactions, while a formally defined operator performs *global* strength propagation. This aligns with work on inferring attack relations for gradual semantics (Oren & Yun, 2023) and broader neural–symbolic approaches (Yu et al., 2025).

**Positioning.** Our results show that structural coherence and persuasive effectiveness are distinct, situating GRASP as a tool for analyzing interaction structure rather than modeling persuasion.

## B. Proofs

We begin by recalling that convergence of iterative schemes is most naturally studied through contraction properties of the underlying operator. In particular, if the GRASP operator is a contraction on a suitable domain, then classical fixed-point results guarantee both existence and uniqueness of an equilibrium, as well as convergence from arbitrary initialization.

### B.1. Preliminaries

We provide here basic results, for a modern reference plase refer to (Bullo, 2022).

**Definition B.1** (Metric Space). Let $\mathcal{S}$ be a non-empty set, a map $d : \mathcal{S} \times \mathcal{S} \to \mathbb{R}$ is a metric if (1) $d(x,y) = 0 \iff x = y$, (2) $\forall x, y \in \mathcal{S},\ d(x,y) = d(y,x)$, (3) $\forall x, y, z \in \mathcal{S},\ d(x,y) \leq d(x,z) + d(y,z)$.

**Definition B.2** (Cauchy/Convergent Sequences). Let $\{x_k\}_{k \in \mathbb{N}}$ be a sequence in $(\mathcal{S}, d)$. We call $\{x_k\}_{k \in \mathbb{N}}$ Cauchy if for any $\epsilon > 0$ there exist $k$ such that, for all $h \in \mathbb{N}$ and $i \geq k$, $d(x_i, x_{i+h}) \leq \epsilon$. We call $\{x_k\}_{k \in \mathbb{N}}$ convergent to $x^* \in \mathcal{S}$ if for any $\epsilon > 0$ there exist $k$ such that $d(x_i, x^*) \leq \epsilon$ for all $i \geq k$.

**Definition B.3** (Complete Metric Space). $(\mathcal{S}, d)$ is complete if every Cauchy sequence in $\mathcal{S}$ converges to a point in $\mathcal{S}$. $(\mathbb{R}^d, \|\cdot\|)$ is complete.

**Definition B.4** (Lipschitz Maps, Contraction). Let $(\mathcal{S}, d)$ be a metric space, $G : \mathcal{S} \to \mathcal{S}$ is Lipschitz if there exists a $\ell \geq 0$ (Lipschitz constant for $G$) such that for all $x, y \in X$, $d(G(x), G(y)) \leq \ell d(x,y)$. If $\ell < 1$ is possible, $G$ is called a contraction.

This well-known result is the workhorse of most convergence analyses.

**Theorem B.5** (Banach Contraction Theorem). *Let $(\mathcal{S}, d)$ be a complete metric space, and $G$ a contraction with factor $\ell$. Then, $G$ has a unique fixed point $s^* \in \mathcal{S}$, and the sequence generated by $s_{k+1} = G(s_k)$ converges to $s^*$.*

## B.2. GRASP Convergence

Recall that GRASP operator $G : \mathbb{R}^n \to \mathbb{R}^n$ is defined coordinatewise as

$$G(s)_j = \frac{1 + \beta \sum_k D_{kj} s_k}{1 + \alpha \sum_i W_{ij} s_i}, \tag{5}$$

where $W_{ij}, D_{ij} \geq 0$ for all $i, j \in [n]$.

**Contraction on compact subspaces.** While global contraction on the entire space is often too strong to expect, many nonlinear operators exhibit contraction behavior when restricted to an invariant subset. This motivates analyzing GRASP on a bounded region of interest, corresponding to a meaningful range of argument strengths.

The following lemma formalizes a standard extension of Banach's theorem to such invariant subsets.

**Lemma B.6** (Banach on subsets). *Let $(\mathcal{S}, d)$ be complete, and assume there is a point $s_0$ and a radius $r$ such that $G : \mathcal{S} \to \mathcal{S}$ is a contraction with Lipschitz constant $\ell$ on $B := \{x \in \mathcal{S}, d(x, s_0) \leq r\}$. Assume $d(s_0, G(s_0)) \leq r(1 - \ell)$, then $B$ is invariant under $G$ and the contraction theorem applies to $G$ restricted to $B$.*

*Proof.* It is sufficient to show that, for $x \in B$, $G(x) \in B$. Note that

$$d(G(x), s_0) \leq d(G(x), G(s_0)) + d(G(s_0), s_0)$$
$$\leq \ell d(x, s_0) + (1 - \ell) r \leq r$$

and hence this concludes the proof. $\qquad\square$

**On damping in GRASP.** Since GRASP incorporates a damping step in practice, it is important to understand whether damping alone can induce contraction. The following remark clarifies that damping cannot compensate for a lack of contraction in the base operator itself.

*Remark* B.7 (Damping cannot turn a non-contraction into a contraction). In GRASP, we use damping:

$$s^+ = \alpha s + (1 - \alpha) G(s) =: \hat{G}(s),$$

Doing cannot make $\ell \leq 1$ if it was not previously so. Indeed, on $(\mathbb{R}^d, \| \cdot \|)$, for an arbitrary norm:

$$\|\hat{G}(x) - \hat{G}(y)\| = \|\alpha(x - y) + (1 - \alpha)(G(x) - G(y))\|$$
$$\leq (\alpha + (1 - \alpha)\ell) \|x - y\|.$$

So if $\ell \leq 1$, the new factor is also $\leq 1$. If $\ell \geq 1$, the new factor is also $\geq 1$.

The remark above shows that to successfully characterize the convergence properties of (undamped) GRASP, it is necessary and sufficient to consider the properties of $s \mapsto G(s)$.

**Approach.** Our approach proceeds in two steps. First, we identify a natural bounded subset of $\mathbb{R}^d$ that is invariant under $G$. Second, we establish that $G$ is a contraction on this set under explicit conditions on the interaction matrices.

### B.2.1. INVARIANCE

Throughout this section, we work with the set

$$\mathcal{S} := \{s \in \mathbb{R}^d : \|s - \mathbf{1}\|_\infty \leq 1\},$$

which corresponds to bounded, nonnegative strength vectors centered around the neutral baseline $\mathbf{1} = (1, \ldots, 1)$.

**Lemma B.8.** *Let $x, y \in \mathbb{R}^d$. Then*

$$\|x \odot y\|_\infty \leq \|x\|_\infty \|y\|_\infty.$$

*Proof.* Let $x, y \in \mathbb{R}^d$ and define $(x \odot y)_i := x_i y_i$. Then

$$\|x \odot y\|_\infty = \max_i |x_i y_i|$$
$$\leq \max_i (|x_i| \|y\|_\infty)$$
$$= \|y\|_\infty \max_i |x_i|$$
$$= \|x\|_\infty \|y\|_\infty,$$

where we used the bound $|y_i| \leq \|y\|_\infty$ for all $i$. $\qquad\square$

We now show that $\mathcal{S}$ is invariant under the GRASP operator. This guarantees that once the iteration enters $\mathcal{S}$, it remains there for all subsequent steps.

**Lemma B.9** (Invariance). *Let $\mathcal{S} := \{s \in \mathbb{R}^d, \|s - 1\|_\infty \leq 1\}$ and let $G : \mathcal{S} \to \mathbb{R}^d$ be defined elementwise as*

$$G(s)_i = \frac{1 + \beta(D^\top s)_i}{1 + \alpha(W^\top s)_i},$$

*for matrices $W, D \in \mathbb{R}^{d \times d}$ and scalars $\alpha, \beta \geq 0$. If $W, D$ have non-negative entries and*

$$\alpha \leq \frac{1}{4\|W\|_1}, \qquad \beta \leq \frac{1}{4\|D\|_1},$$

*then $G(\mathcal{S}) \subseteq \mathcal{S}$.*

*Proof.* Using the triangle inequality and denoting $\mathbf{1} = (1, \ldots, 1) \in \mathcal{S}$, we obtain

$$\|G(s) - \mathbf{1}\|_\infty = \left\| \frac{1 + \beta D^\top s}{1 + \alpha W^\top s} - \mathbf{1} \right\|_\infty$$
$$= \left\| \frac{(\beta D^\top - \alpha W^\top) s}{1 + \alpha W^\top s} \right\|_\infty.$$

Since $\alpha > 0$ and $W$ has non-negative entries, the denominator satisfies $1 + \alpha W^\top s > 1$ elementwise and can be dropped for an upper bound. Using norm subadditivity,

$$\begin{aligned} \|G(s) - \mathbf{1}\|_\infty &\leq \|(\beta D^\top - \alpha W^\top)s\|_\infty \\ &\leq \|\beta D^\top - \alpha W^\top\|_\infty \|s\|_\infty \\ &\leq (\beta\|D^\top\|_\infty + \alpha\|W^\top\|_\infty)\|s\|_\infty. \end{aligned}$$

Using the identity $\|M^\top\|_\infty = \|M\|_1$ and the fact that $s \in \mathcal{S}$ implies $\|s\|_\infty \leq 2$, we conclude

$$\|G(s) - \mathbf{1}\|_\infty \leq (\beta\|D\|_1 + \alpha\|W\|_1) \cdot 2 \leq \left(\tfrac{1}{4} + \tfrac{1}{4}\right) \cdot 2 = 1.$$

$\square$

### B.2.2. LIPSCHITZ CONSTANT

We next establish a Lipschitz bound for $G$ on $\mathcal{S}$, which will allow us to invoke Banach's fixed-point theorem.

**Lemma B.10** (Lipschitz Constant). *Consider the GRASP operator defined coordinatewise as*

$$G(s)_i = \frac{1 + \beta(D^\top s)_i}{1 + \alpha(W^\top s)_i}.$$

*Let $x, y \in \mathcal{S}$. If $W, D$ have non-negative entries, then*

$$\|G(x) - G(y)\|_\infty \leq \ell\|x - y\|_\infty,$$

$$\ell := \beta\|D\|_1 + \alpha\|W\|_1 \frac{\|G(x)\|_\infty + \|G(y)\|_\infty}{2}.$$

*Proof.* Define elementwise

$$(\mathcal{N}_x)_i := 1 + \beta(D^\top x)_i, \qquad (\mathcal{D}_x)_i := 1 + \alpha(W^\top x)_i.$$

Then

$$\begin{aligned} \|G(x) - G(y)\|_\infty &= \left\|\frac{\mathcal{N}_x}{\mathcal{D}_x} - \frac{\mathcal{N}_y}{\mathcal{D}_y}\right\|_\infty \\ &= \left\|\frac{\mathcal{N}_x \mathcal{D}_y - \mathcal{N}_y \mathcal{D}_x}{\mathcal{D}_x \mathcal{D}_y}\right\|_\infty. \end{aligned}$$

Expanding the numerator in two symmetric ways,

$$\mathcal{N}_x \mathcal{D}_y - \mathcal{N}_y \mathcal{D}_x = (\mathcal{N}_x - \mathcal{N}_y)\mathcal{D}_y + (\mathcal{D}_y - \mathcal{D}_x)\mathcal{N}_y,$$
$$\mathcal{N}_x \mathcal{D}_y - \mathcal{N}_y \mathcal{D}_x = (\mathcal{N}_x - \mathcal{N}_y)\mathcal{D}_x + (\mathcal{D}_y - \mathcal{D}_x)\mathcal{N}_x.$$

Averaging these expressions yields

$$|G(x) - G(y)\|_\infty$$
$$= \left\|(\mathcal{N}_x - \mathcal{N}_y)\frac{\mathcal{D}_x + \mathcal{D}_y}{2\mathcal{D}_x \mathcal{D}_y} + (\mathcal{D}_y - \mathcal{D}_x)\frac{\mathcal{N}_x + \mathcal{N}_y}{2\mathcal{D}_x \mathcal{D}_y}\right\|_\infty$$

Applying Lemma B.8 and the triangle inequality,

$$\begin{aligned} \|G(x) - G(y)\|_\infty \leq{}& \|\mathcal{N}_x - \mathcal{N}_y\|_\infty \left\|\frac{\mathcal{D}_x + \mathcal{D}_y}{2\mathcal{D}_x \mathcal{D}_y}\right\|_\infty \\ &+ \|\mathcal{D}_y - \mathcal{D}_x\|_\infty \left\|\frac{\mathcal{N}_x + \mathcal{N}_y}{2\mathcal{D}_x \mathcal{D}_y}\right\|_\infty. \end{aligned}$$

Since $\mathcal{D}_x, \mathcal{D}_y \geq 1$ elementwise,

$$\left\|\frac{\mathcal{D}_x + \mathcal{D}_y}{2\mathcal{D}_x \mathcal{D}_y}\right\|_\infty \leq 1.$$

Moreover,

$$\begin{aligned} \left\|\frac{\mathcal{N}_x + \mathcal{N}_y}{2\mathcal{D}_x \mathcal{D}_y}\right\|_\infty &\leq \frac{1}{2}\left(\left\|\frac{\mathcal{N}_x}{\mathcal{D}_x}\right\|_\infty + \left\|\frac{\mathcal{N}_y}{\mathcal{D}_y}\right\|_\infty\right) \\ &= \frac{\|G(x)\|_\infty + \|G(y)\|_\infty}{2}. \end{aligned}$$

Finally,

$$\|\mathcal{N}_x - \mathcal{N}_y\|_\infty \leq \beta\|D^\top\|_\infty \|x - y\|_\infty = \beta\|D\|_1\|x - y\|_\infty,$$

and similarly

$$\|\mathcal{D}_x - \mathcal{D}_y\|_\infty \leq \alpha\|W\|_1\|x - y\|_\infty.$$

$\square$

The convergence result in the main paper naturally follows.

### B.2.3. PROOF OF THEOREM 3.1

*Proof.* The first part is exactly Lemma B.9. The second follows from Lemma B.10, since under the assumptions on $\alpha, \beta$ we have

$$\begin{aligned} \ell &= \max_{x,y \in \mathcal{S}}\left[\beta\|D\|_1 + \alpha\|W\|_1 \frac{\|G(x)\|_\infty + \|G(y)\|_\infty}{2}\right] \\ &\leq \beta\|D\|_1 + 2\alpha\|W\|_1 \leq \frac{1}{4} + \frac{1}{2} < 1, \end{aligned}$$

where we used that $\|G(s)\|_\infty \leq 2$ for all $s \in \mathcal{S}$. $\square$

## C. Illustrative Example: Dynamic Ranking Shift and Convergence

This scenario demonstrates how GRASP can exhibit *transient rank reversals* before converging to a stable ordering, even in a small graph.

**Setup.** We consider four arguments:

- $a_1$: The central claim, attacked by $a_3$.

- $a_2$: A strong, unattacked argument that weakly attacks $a_4$.

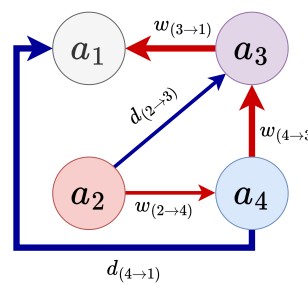

*Figure 5.* Argumentation graph for the dynamic example. The "hero" argument $\mathbf{a_4}$ attacks $\mathbf{a_3}$, thereby defending the initial claim $\mathbf{a_1}$. However, the independent argument $\mathbf{a_2}$ attacks $\mathbf{a_4}$, which indirectly restores strength to the attacker $\mathbf{a_3}$. This interaction induces a non-trivial reordering of argument strengths over GRASP iterations.

- $a_3$: A strong attacker of $a_1$, itself attacked by $a_4$.

- $a_4$: A "hero" argument that attacks $a_3$, thereby defending $a_1$.

The weighted attack matrix $W$ (rows = attackers, columns = targets) is

$$W = \begin{bmatrix} 0 & 0 & 0 & 0 \\ 0 & 0 & 0 & 0.3 \\ 1 & 0 & 0 & 0 \\ 0 & 0 & 1 & 0 \end{bmatrix},$$

corresponding to

$$w_{31} = 1.0, \qquad w_{43} = 1.0, \qquad w_{24} = 0.3.$$

The second-order defense matrix is

$$D = WW = \begin{bmatrix} 0 & 0 & 0 & 0 \\ 0.3 & 0 & 0 & 0 \\ 0 & 0 & 0 & 0 \\ 1 & 0 & 0 & 0 \end{bmatrix},$$

yielding

$$d_{41} = 1.0 \quad (a_4 \text{ defends } a_1),$$
$$d_{23} = 0.3 \quad (a_2 \text{ weakly defends } a_3).$$

We use the (undamped) GRASP operator

$$F_j(\mathbf{s}) \;=\; \frac{1 + \beta(D^\top \mathbf{s})_j}{1 + \alpha(W^\top \mathbf{s})_j}, \qquad \mathbf{s}^{(t+1)} = F(\mathbf{s}^{(t)}),$$

with $\alpha = 1.0$ and $\beta = 0.5$, and initialize

$$\mathbf{s}^{(0)} = [1, 1, 1, 1]^\top.$$

**Iteration 1.**

$$G^{(0)} = [0.75, \; 1.00, \; 0.58, \; 0.77]$$

$$\mathbf{s}^{(1)} = G^{(0)}.$$

**Ranking 1:** $a_2 > a_4 > a_1 > a_3$.

*Observation.* $a_2$ is strongest because it has no attackers. $a_3$ is heavily penalized by $a_4$'s attack, while $a_1$ is still strongly suppressed by $a_3$.

**Iteration 2.**

$$G^{(1)} = [0.81, \; 1.00, \; 0.61, \; 0.77]$$

$$\mathbf{s}^{(2)} = G^{(1)}.$$

**Ranking 2:** $a_2 > a_1 > a_4 > a_3$    **(Rank change)**

*Observation.* As $a_4$ continues to weaken $a_3$, pressure on $a_1$ decreases, allowing $a_1$ to recover. At the same time, the weak but persistent attack from $a_2$ slowly reduces $a_4$'s score, enabling $a_1$ to overtake $a_4$.

**Iteration 3 (Convergence).**

$$G^{(2)} = [0.83, \; 1.00, \; 0.63, \; 0.77]$$

$$\mathbf{s}^{(3)} = G^{(2)}.$$

**Ranking 3:** $a_2 > a_1 > a_4 > a_3$.

The ordering is now stable.

**Ranking Summary.**

$$\text{Iteration 1: } a_2 > a_4 > a_1 > a_3$$
$$\text{Iteration 2: } a_2 > a_1 > a_4 > a_3$$
$$\text{Iteration 3: } a_2 > a_1 > a_4 > a_3$$

**Takeaway.** This example highlights a key property of GRASP: rankings are not determined solely by immediate attacks, but by the evolving balance between attacks and counter-attacks. Temporary rank reversals naturally arise as indirect defenses accumulate, and convergence reflects a stable global equilibrium of structural influence.

## D. Structural Evaluation on Synthetic Graphs

To evaluate whether ranking operators respect the principles of *structural sufficiency* (Section 4), we construct a controlled synthetic testbed of argumentation graphs with explicitly defined interaction structure. Unlike natural debates, these graphs admit unambiguous structural ground truth, allowing direct evaluation against *necessary ranking*

*constraints* derived from structural sufficiency rather than subjective judgment.

Our objective is not to induce a total order, but to test whether a method violates any ranking relations that must hold whenever an argument is structurally sufficient.

### D.1. Structural Archetypes

We consider a suite of small, canonical *structural archetypes* as demonstrated in Fig.!6, each isolating a distinct dialectical phenomenon. Each instance is a directed argumentation graph $\mathcal{G} = (A, R^-)$ with $|A| \in [4, 6]$. Edges represent attacks with positive weight; self-attacks are disallowed. To avoid degenerate sparsity, we optionally add low-weight noise attacks that preserve the core structure while increasing heterogeneity.

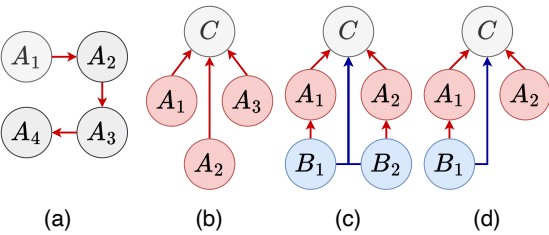

|(a)|(b)|(c)|(d)|

*Figure 6.* Canonical structural archetypes used in synthetic evaluation. (a) **Attack Chain (Reinstatement):** linear chain of attacks where counter-attacks reinstate upstream arguments. (b) **Fork (Convergent Attack):** multiple arguments attacking a single target. (c) **Diamond (Cascading Defense):** parallel attackers of a claim followed by a convergent counter-attack. (d) **Bipolar Structure:** multiple attackers of a claim with a downstream attack on one attacker.

- **Attack Chain (Reinstatement):** $a_1 \to a_2 \to \cdots \to a_n$, testing whether counter-attacks reinstate downstream arguments.

- **Fork (Convergent Attack):** Multiple arguments attacking a single target.

- **Diamond (Cascading Defense):** Parallel attacks followed by convergent counter-attacks.

- **Bipolar Structure:** Multiple attackers of a claim followed by a downstream attack.

**Random DAG Stress Tests.** In addition to canonical motifs, we include random directed acyclic graphs (DAGs) with $n = 20$ nodes and edge probabilities $p \in \{0.1, 0.3\}$. Edge weights are sampled uniformly from $[0.2, 1.0]$. These graphs do not encode specific dialectical motifs and are used solely to test robustness and convergence under heterogeneous structure.

### D.2. Critical Ranking Conditions (CRCs)

For each archetype, we derive *critical ranking conditions* (CRCs), expressed as pairwise constraints $a \succ b$. CRCs are *necessary conditions only* implied by structural sufficiency and intentionally do not define a total order.

Each CRC is grounded in a specific axiom:

- **Attack Sensitivity (S1):** In Fork structures, a target with unneutralized attackers must not outrank its attackers.

- **Defense Reinstatement (S2):** In Attack Chain and Diamond motifs, an argument whose attacker is itself attacked must outrank the original attacker.

- **Structural Locality (S3):** In Bipolar structures, arguments not structurally connected to a claim must not affect its ranking.

- **Baseline Sufficiency (S4):** Unattacked arguments must not be ranked below attacked ones.

For random DAGs, only *baseline sufficiency* (S5) constraints are imposed. Violations on these graphs therefore reflect robustness limitations under noisy, non-motif structure rather than axiomatic failure.

CRCs deliberately exclude monotonic or comparative claims (e.g., "adding support must increase strength"), which are not implied by structural sufficiency.

### D.3. Methods and Metrics

We compare GRASP against standard structural ranking baselines. All methods operate solely on the weighted attack matrix $W \in \mathbb{R}_{\geq 0}^{n \times n}$, where $W_{ij}$ denotes the strength of the attack from argument $a_i$ to argument $a_j$. Each method produces a real-valued *strength score* $s_j \in \mathbb{R}_{>0}$ for every argument $a_j$; final rankings are obtained by ordering arguments in decreasing $s_j$.

**H-Categorizer.** The H-categorizer (Besnard & Hunter, 2001) penalizes arguments proportionally to the total strength of their attackers: $s_j = \frac{1}{1+\sum_i W_{ij}}$. This method is static and purely local, ignoring defense and higher-order structure.

**KatzAttack.** We adapt Katz centrality (Katz, 1953) to attack graphs by accumulating all discounted attack paths: $c = (I - \lambda W^\top)^{-1}\mathbf{1}$, with $s_j = 1/c_j$. Here $\lambda > 0$ is a damping parameter controlling the contribution of longer attack paths; it is chosen sufficiently small to ensure convergence. Inverting $c_j$ ensures that arguments receiving many (direct or indirect) attacks receive lower strength.

| Method | Viol. | Sev. | Iter. | Conv. |
|---|---|---|---|---|
| GRASP | **0.003** | **0.010** | 65.8 | 100% |
| GRASP ($\beta = 0$) | 0.071 | 0.019 | 63.7 | 100% |
| Defense Ratio | 0.042 | 0.042 | – | – |
| KatzAttack | 0.228 | 0.012 | – | – |
| H-Categorizer | 0.228 | 0.042 | – | – |
| Binary Indegree | 0.233 | 0.031 | – | – |
| Max Incoming Attack | 0.290 | 0.013 | – | – |

*Table 4.* Structural evaluation on synthetic graphs. Violation rate and severity are measured against critical ranking conditions derived from structural sufficiency. Lower is better.

| Method | Mean $\rho$ | Median $\rho$ |
|---|---|---|
| **Multi-turn** | | |
| GRASP-$W_1$ | -0.998 | -0.998 |
| GRASP-$W_\infty + \bar{D}$ | -0.994 | -0.997 |
| GRASP-$W_1 + \bar{D}$ | -0.982 | -0.988 |
| GRASP | -0.960 | -0.973 |
| GRASP-$W_\infty$ | -0.958 | -0.973 |
| RAW | 0.071 | 0.093 |
| **Pool** | | |
| GRASP-$W_1$ | -1.000 | -1.000 |
| GRASP-$W_\infty + \bar{D}$ | -0.996 | -0.997 |
| GRASP-$W_1 + \bar{D}$ | -0.986 | -0.990 |
| GRASP | -0.955 | -0.967 |
| GRASP-$W_\infty$ | -0.954 | -0.966 |
| RAW | -0.007 | 0.049 |

*Table 5.* Spearman correlation between rankings and in-strength centrality. Strong negative values indicate that highly-attacked arguments are ranked lower.

**Defense Ratio.** As a non-iterative defense-aware baseline, we evaluate a closed-form ratio of total two-hop defense to total incoming attack $s_j = \frac{1 + \sum_k (W^2)_{kj}}{1 + \sum_i W_{ij}}$. This aggregates defense measured by length-two attack paths but does not perform iterative strength propagation.

**Binary Indegree.** Counts the number of distinct attackers, ignoring attack magnitude: $s_j = \frac{1}{1 + \sum_i \mathbb{I}[W_{ij} > 0]}$.

**Max Incoming Attack.** penalizes an argument by the strength of its strongest attacker: $s_j = \frac{1}{1 + \max_i W_{ij}}$.

**Metrics.** Given a set of critical ranking conditions (CRCs) of the form $a \succ b$, a violation occurs whenever $s_a \leq s_b$. We report: (i) **violation rate** (fraction of violated CRCs), (ii) **violation severity** (mean normalized margin $s_b - s_a$ over violations), (iii) **mean iterations** to convergence (iterative methods only), and (iv) **convergence fraction**. Fraction of graphs on which the iterative method converged in finite steps.

### D.4. Results

Table 4 summarizes performance across all archetypes and random DAG stress tests.

**Summary.** GRASP achieves the lowest violation rate across the synthetic suite and satisfies all CRCs derived from canonical structural archetypes. The few remaining violations arise exclusively in random DAG stress tests and are caused by low-weight noise attacks that induce weak, competing attack paths without clear dialectical resolution. Disabling defense propagation ($\beta = 0$) substantially increases violations, confirming that higher-order structural interaction is essential. Overall, these results demonstrate that structural sufficiency cannot be captured by local, linear, or non-propagative aggregation alone.

## E. Additional Results

### E.1. Centrality Alignment and Structural Centers

We test whether argument rankings align with simple graph-theoretic notions of centrality derived from the attack matrix $W$. For each debate and NLI model, we construct a weighted directed graph with edge weights given by contradiction probabilities and compute standard centralities: in-strength, out-strength, net-strength, and PageRank. Across all settings, *in-strength* (total incoming attack mass) provides the strongest and most stable signal, and we therefore report results using this centrality only.

**Ranking–centrality alignment.** Table 5 reports Spearman correlation between each ranking and the in-strength ordering. All GRASP variants exhibit extremely strong negative correlation (median $\rho \approx -0.96$ to $-0.99$), indicating that arguments receiving more attacks are consistently ranked lower. In contrast, RAW LLM rankings show near-zero correlation, demonstrating little sensitivity to structural position.

**Existence of a structural center.** We ask whether independently constructed attack graphs (from different NLI judge models) agree on which argument is structurally most central. For each debate and NLI model, we compute the in-strength central argument (top-1 by incoming attack mass), and measure the fraction of models that select the same argument.

**Takeaway.** Attack graphs induced by different NLI models exhibit a moderately stable structural center. Combined with the strong alignment between GRASP rankings and in-strength (Table 5), this suggests that GRASP recovers a

| Setting | Mean vote | Median | 90th pct |
|---|---|---|---|
| Multi-turn | 0.526 | 0.500 | 0.833 |
| Pool | 0.463 | 0.417 | 0.667 |

*Table 6.* Consensus over the most central argument (in-strength) across NLI models. Values indicate the fraction of models selecting the same top-1 argument.

shared structural signal present in the attack graphs, whereas RAW LLM rankings do not consistently track this structure.

### E.2. Ablation: Structural Similarity vs. Ranking Similarity

We study whether similarity between induced attack graphs translates into similarity of final GRASP rankings. For each pair of NLI judge models $(m_i, m_j)$, and for each setting (pool or multi-turn), we compute:

- The mean Pearson correlation between their attack matrices, $\rho(W^{(i)}, W^{(j)})$, using vectorized off-diagonal entries.

- The mean Kendall $\tau$ correlation between the corresponding GRASP final rankings produced under the two judges, averaged across debates.

Each point in Figure 7 corresponds to one unordered pair of judge models. The horizontal axis measures similarity of the induced attack graphs, while the vertical axis measures similarity of the resulting GRASP rankings.

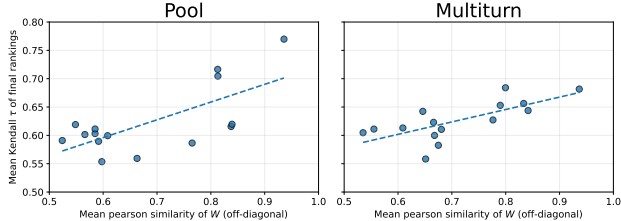

*Figure 7.* Relationship between similarity of attack graphs ($W$) and similarity of final GRASP rankings. Left: pool setting. Right: multi-turn setting. Dashed lines show least-squares linear fits.

Across both settings, we observe a clear positive association: pairs of judges whose induced attack matrices are more strongly correlated tend to produce more similar GRASP rankings. The trend is consistent in both pool and multi-turn regimes, though with non-negligible dispersion, indicating that while graph similarity is not the sole determinant of ranking behavior, it explains a meaningful portion of inter-model agreement.

**Takeaway.** Agreement at the level of local pairwise attack judgments propagates to agreement in global GRASP rankings. This supports the view that GRASP's stability arises

| Method | Angle | Top-3 | Top-5 | SwapDist | Spearman $\rho$ |
|---|---|---|---|---|---|
| | | **GRASP** | | | |
| GRASP | Social | 0.888 | 0.984 | 0.166 | 0.746 |
| GRASP | Legal | 0.885 | 0.982 | 0.180 | 0.724 |
| GRASP | Moral | 0.862 | 0.977 | 0.175 | 0.738 |
| GRASP | Economic | 0.857 | 0.976 | 0.187 | 0.705 |
| GRASP | Political | 0.865 | 0.908 | 0.162 | 0.764 |
| GRASP | Technological | 0.844 | 0.904 | 0.171 | 0.758 |
| | | **RAW** | | | |
| RAW | Social | 0.816 | 0.980 | 0.309 | 0.421 |
| RAW | Legal | 0.794 | 0.978 | 0.342 | 0.352 |
| RAW | Moral | 0.801 | 0.963 | 0.310 | 0.419 |
| RAW | Economic | 0.795 | 0.958 | 0.320 | 0.399 |
| RAW | Political | 0.665 | 0.861 | 0.297 | 0.469 |
| RAW | Technological | 0.632 | 0.848 | 0.346 | 0.359 |

*Table 7.* Angle-level inter-model agreement, averaged across all debates and settings. GRASP variants consistently yield higher agreement than RAW across all semantic angles.

primarily from the shared geometry of the induced attack graphs, rather than from properties of any specific judge model or from optimization artifacts.

### E.3. Angle-Level Agreement Analysis

We further analyze inter-model agreement at the level of *semantic angles* (ECONOMIC, LEGAL, MORAL, POLITICAL, SOCIAL, TECHNOLOGICAL). For each debate and angle, we restrict each model's ranking to the subset of arguments belonging to that angle and compute pairwise agreement between all model pairs. Metrics are then averaged within debate and finally aggregated across all debates from both pool and multi-turn settings.

Table 7 reports mean Top-$k$ overlap, normalized swap distance, and Spearman's $\rho$ for RAW rankings and each GRASP variant.

**Key observations.** Across all angles, GRASP variants exhibit substantially higher agreement than RAW rankings. Differences between GRASP variants are small, indicating that agreement gains are driven primarily by structural aggregation rather than the specific normalization choice. The pattern is consistent across angles, showing that GRASP's benefits are not confined to any particular argumentative dimension.

## F. Debate Motions

We list the 50 debate motions used in STRUCTDEBATE, grouped by thematic category.

### Technology & AI

1. This House would ban the use of AI in primary and secondary education.

2. This House would ban stablecoins pegged to national currencies.

3. This House would mandate all businesses to accept only digital payments.

4. This House would require electric vehicle manufacturers to refuse sales in countries with poor environmental records.

5. This House would allow individuals to erase morally distressing memories.

6. This House would ban facial recognition technology in public spaces.

7. This House would require social media companies to make their recommendation algorithms public.

## Economics & Labor

8. This House would abolish the minimum wage law.

9. This House would allow the sale and purchase of human organs.

10. This House would ban sovereign wealth funds from investing in private equity.

11. This House would require companies to make the salaries of all their employees publicly available.

12. This House would allow workers in less economically developed countries to waive labor protections in exchange for higher wages.

13. This House would allow lump-sum scholarships as an alternative to periodic disbursements.

14. This House would introduce a universal basic income funded by wealth taxes.

## Law, Rights, and Governance

15. This House would require warrants for searches instead of allowing stop-and-frisk.

16. This House would criminalize dangerous in-play actions in professional sport.

17. This House would ban corporate donations to political campaigns.

18. This House would allow democratic governments to overturn supranational court decisions with a simple legislative majority.

19. This House would introduce a binding "None of the Above" option on national election ballots.

20. This House would allow citizens to vote directly on impeachment cases through a national referendum.

21. This House would allow subnational jurisdictions to overturn federal policies via citizen referendum.

## Social & Moral Policy

22. This House would allow parents to administer behavioral enhancement drugs to children without their consent.

23. This House would mandate comprehensive queer education in schools.

24. This House would restrict state funding only to art perceived as valuable by the general public.

25. This House would ban the private ownership of historical artifacts.

26. This House would allow high school students to rate teachers as a primary basis for pay increases.

27. This House would introduce compulsory national service for all citizens.

## Environment & Development

28. This House would invest preferentially in climate startups in developing countries rather than developed countries.

29. This House would ban the export of waste to developing countries.

30. This House would nationalize luxury ingredient production in producing countries.

31. This House would impose carbon tariffs on imported goods.

32. This House would ban advertising for environmentally harmful products.

## International Relations & Identity

33. This House would ban countries from offering financial incentives to foreign athletes to switch nationality.

34. This House would allow prisoners serving life without parole to opt for the death penalty.

35. This House would ban proselytization acts in liberal democracies.

36. This House would restrict immigration based on environmental carrying capacity.

37. This House would allow refugees to be settled through private sponsorship markets.

## Media, Culture, and Education

38. This House would ban the consolidation of major news organizations.

39. This House would introduce a youth-weighted voting system in democratic elections.

40. This House would require public broadcasters to allocate equal airtime to all political parties.

41. This House would abolish standardized testing in university admissions.

42. This House would mandate media literacy education for all adults.

## Security & State Power

43. This House would ban private military contractors.

44. This House would allow preventive detention for credible terrorism threats.

45. This House would restrict police use of lethal force to extreme circumstances only.

## Science & Bioethics

46. This House would allow gene editing of embryos for non-medical traits.

47. This House would require mandatory vaccination for all citizens.

48. This House would ban animal testing for cosmetic products.

## Platform Power & Markets

49. This House would break up dominant technology monopolies.

50. This House would require platforms to compensate users for personal data usage.

## G. Hyperparameter Selection via Cross-Model Agreement

We perform a *post-hoc* grid search to analyze the sensitivity of GRASP to its hyperparameters, using *cross-model agreement* among GRASP-induced rankings as the diagnostic signal. Intuitively, if GRASP captures a stable structural signal, then different NLI judges used to construct the attack graph $W$ should yield highly similar final rankings under a broad range of hyperparameter choices. Importantly, this analysis is purely diagnostic: all main experiments use a single set of *a priori* fixed hyperparameters, and the grid search is used only to assess robustness rather than to tune the reported results.

**Protocol.** For a given triple $(\alpha, \beta, \gamma)$, we compute a GRASP ranking for each NLI model and measure the mean pairwise Kendall-$\tau$ agreement between all model pairs. We then average this agreement across debates.

We perform a grid search over

$$\alpha \in \{0.1, 0.25, 0.5, 1.0\},$$
$$\beta \in \{0.1, 0.25, 0.5, 0.75\},$$
$$\gamma \in \{0.6, 0.8, 0.9, 1.0\}.$$

**Results.** The best-performing configuration is:

$$\alpha = 1.0, \qquad \beta = 0.25, \qquad \gamma = 0.6,$$

achieving a mean pairwise Kendall-$\tau$ agreement of 0.624 across 300 debates.

**Takeaway.** We fix $\alpha = 1.0$, $\beta = 0.25$, and $\gamma = 0.6$ for all experiments. More importantly, the high agreement level ($\approx 0.62$ Kendall-$\tau$) demonstrates that GRASP produces highly consistent structural rankings across independently constructed attack graphs, supporting the claim that it extracts a model-agnostic structural signal rather than overfitting to any single NLI judge.

## H. Prompts and Prompt Optimization

### H.1. Prompt Optimization for RAW Rankings

We study whether refining the RAW ranking prompt can improve inter-judge agreement. All prompts are evaluated with temperature 0. We compare three variants: (i) the original RAW prompt used in the main paper (Table 2, denoted RAW), (ii) a more explicit adjudication prompt emphasizing logic, impact, and relevance (RAW-v1, prompt H.2), and (iii) a comparative ranking prompt emphasizing supersession and logical dominance (RAW-v2, prompt H.2).

For each prompt, we recompute inter-model agreement across judges and report mean Kendall's $\tau$, mean Kendall swap distance, top-$k$ overlap, and Spearman correlation. Table 8 summarizes results and reports deltas in parentheses relative to the original RAW prompt.

**Discussion.** Neither refined prompt improves Kendall agreement or Spearman correlation over the original RAW prompt. Both RAW-v2 and RAW-v3 substantially reduce Kendall's $\tau$ in both settings ($\approx 0.14$–$0.20$ absolute drop), indicating even weaker global consistency across judges.

| Method | $\tau$ | Swap | $\rho$ | Top-3 | Top-5 |
|---|---|---|---|---|---|
| **Multi-turn Setting** | | | | | |
| RAW (original) | 0.309 | 0.345 | 0.380 | 0.410 | 0.474 |
| RAW-v2 | 0.147 | 0.426 | 0.193 | 0.507 | 0.565 |
| RAW-v3 | 0.114 | 0.443 | 0.156 | 0.410 | 0.477 |
| **Pool Setting** | | | | | |
| RAW (original) | 0.337 | 0.331 | 0.425 | 0.385 | 0.417 |
| RAW-v2 | 0.175 | 0.413 | 0.229 | 0.402 | 0.463 |
| RAW-v3 | 0.172 | 0.414 | 0.235 | 0.382 | 0.436 |

*Table 8.* Effect of RAW prompt optimization on inter-model agreement. Swap denotes *normalized* Kendall swap distance, computed as $(1 - \tau)/2$.

While small gains appear in top-$k$ overlap, they coincide with increases in swap distance, signaling that superficial agreement among a few top arguments masks large-scale ranking instability.

**Takeaway.** Prompt engineering alone does not make end-to-end RAW ranking reliable. Even carefully structured adjudication instructions fail to recover high inter-model agreement. This reinforces the core claim of the paper: RAW ranking is intrinsically brittle as a global task, whereas GRASP achieves robustness by deriving rankings from more reliable local pairwise interactions.

### H.2. Prompt Schemas

Below we provide the exact prompt schemas used in our experiments. All generations were sampled with temperature 0.

**POOL argument generation prompt.**

```
POOL_SYSTEM = "You generate debate arguments. Output
↪   must be valid JSON only."

def pool_user_content(motion, side, angle, k):
    return json.dumps({
        "task": "Generate short debate arguments.",
        "motion": motion,
        "side": side,
        "angle": angle,
        "num_arguments": k,
        "constraints": {
            "length": "2-3 sentences each",
            "style": "plain, analytical, no
↪               rhetorical flourish",
            "no_lists": True,
            "no_citations": True,
            "no_quotes": True,
            "one_core_claim_plus_one_reason": True,
            "avoid_metaphor": True
        },
        "output_requirements": [
            "Return ONLY a JSON object.",
            "No markdown, no code fences, no
↪               commentary.",
            "Schema: {\"arguments\": [\"...\",
↪               \"...\"]}"
```

```
        ]
    }, ensure_ascii=False)
```

**MULTITURN argument generation prompt.**

```
MULTITURN_SYSTEM = "You participate in a structured
↪   debate. Output must be valid JSON only."

def multiturn_user_content(motion, side, angle,
↪   history, turn_idx):
    return json.dumps({
        "task": "Write one debate turn.",
        "motion": motion,
        "side": side,
        "required_angle": angle,
        "turn_index": turn_idx,
        "debate_history": history,
        "constraints": {
            "length": "2-4 sentences",
            "must_address_previous": (side ==
↪               "Con"),
            "style": "plain, analytical, no
↪               rhetorical flourish",
            "no_lists": True,
            "no_citations": True,
            "no_quotes": True,
            "avoid_metaphor": True
        },
        "output_requirements": [
            "Return ONLY a JSON object.",
            "No markdown, no code fences, no
↪               commentary.",
            "Schema: {\"text\": \"...\"}"
        ]
    }, ensure_ascii=False)
```

**RAW ranking prompt (Original).**

```
RAW_SYSTEM = (
    "You are a careful debate judge. "
    "Rank arguments by how strong and sufficient
↪       they are."
    "Return ONLY valid JSON."
)

def raw_user_payload(motion: str, args: list[dict]):
    return {
        "task": "Rank debate arguments by structural
↪           strength for the motion.",
        "motion": motion,
        "arguments": [
            {
                "id": a["arg_id"],
                "side": a["side"],
                "angle": a["angle"],
                "turn": int(a["turn"]),
                "text": a["text_trunc"],
            }
            for a in args
        ],
        "output_requirements": [
            "Return ONLY a JSON object.",
            "No markdown, no code fences, no
↪               commentary.",
            "Schema: {\"ranking\": [\"<arg_id>\",
↪               ...]}",
            "ranking must contain each input id
↪               exactly once."
        ],
    }
```

**RAW ranking prompt (v2).**

```
RAW_SYSTEM = (
    "You are an expert World Schools Debate
    ↪  adjudicator. "
    "Your goal is to evaluate arguments based on
    ↪  logical coherence, evidence, and impact. "
    "You must remain neutral and ignore your own
    ↪  stance on the motion. "
    "Return ONLY valid JSON."
)

def raw_user_payload(motion: str, args: list[dict]):
    return {
        "task": "Rank the provided debate arguments
        ↪  from strongest to weakest.",
        "motion": motion,
        "evaluation_criteria": {
            "1. Logic": "Are the premises true and
            ↪  does the conclusion follow? Is the
            ↪  reasoning explained clearly?",
            "2. Impact": "Does the argument show why
            ↪  this outcome matters significantly
            ↪  to the stakeholders?",
            "3. Relevance": "How directly does it
            ↪  address the specific motion
            ↪  provided?"
        },
        "arguments": [
            {
                "id": a["arg_id"],
                "side": a["side"],
                "text": a["text_trunc"],
            }
            for a in args
        ],
        "output_requirements": [
            "Return ONLY a raw JSON object.",
            "DO NOT include markdown formatting.",
            "Schema: {\"ranking\":
            ↪  [\"<arg_id_best>\", ...]}",
            "ranking must contain every input id
            ↪  exactly once."
        ],
    }
```

**RAW ranking prompt (v3).**

```
RAW_SYSTEM = (
    "You are a rigorous logic engine designed to
    ↪  compare debating points. "
    "Determine which arguments successfully
    ↪  supersede or outweigh the others. "
    "Return ONLY valid JSON."
)

def raw_user_payload(motion: str, args: list[dict]):
    return {
        "task": "Perform a comparative ranking of
        ↪  the debate arguments for the given
        ↪  motion.",
        "motion": motion,
        "instructions": [
            "Read all arguments first.",
            "Identify arguments that rely on logical
            ↪  fallacies and rank them lower.",
            "Identify arguments with strong
            ↪  mechanisms and high-stakes impacts
            ↪  and rank them higher.",
            "If two arguments are similar, rank the
            ↪  one with more nuance/detail higher."
        ],
        "arguments": [
            {
                "id": a["arg_id"],
                "side": a["side"],
                "angle": a["angle"],  # Angle is
                ↪  useful context for comparison
                "text": a["text_trunc"],
```

```
            }
            for a in args
        ],
        "output_requirements": [
            "Output valid JSON only.",
            "No prologue or epilogue.",
            "Schema: {\"ranking\":
            ↪  [\"<strongest_arg_id>\", ...,
            ↪  \"<weakest_arg_id>\"]}",
            "Ensure strict adherence to the schema."
        ],
    }
```

**NLI attack-scoring prompt used to construct $W$ (GRASP).**

```
NLI_SYSTEM = (
    "You are an interaction scorer for debate
    ↪  arguments.\n"
    "Given Text A and Text B, output how strongly B
    ↪  attacks/undermines A.\n\n"
    "Return ONLY valid JSON with keys:\n"
    "{\"attack_score\": number}\n\n"
    "Rules:\n"
    "- attack_score must be a real-valued number in
    ↪  [0,1].\n"
    "- 0.0 means: B does not undermine A at all
    ↪  (supportive or unrelated).\n"
    "- 1.0 means: B directly contradicts or strongly
    ↪  refutes A.\n"
    "- Use the full continuous range; do NOT
    ↪  restrict to discrete steps.\n"
    "- You must always output a score (never
    ↪  null).\n"
    "- Output JSON only. No extra text."
)

def nli_user_payload(text_a: str, text_b: str):
    return {
        "task": "Directed attack scoring",
        "text_a": text_a,
        "text_b": text_b,
        "output_format": {"attack_score": "float in
        ↪  [0,1]"},
        "rules": ["Output JSON only. No markdown. No
        ↪  extra keys."],
    }
```

# I. Case Study Argument Texts

Table 9 lists the arguments corresponding to the volatile trajectories shown in Figure 4 for the debate `mt_048_x-ai__grok-4` ("This House would break up dominant technology monopolies."). Arguments are ordered by the labels used in the figure.

# J. Attack-Graph Geometry Across Debates (Qualitative Appendix)

We provide qualitative visualizations of induced attack graphs for four multi-turn debates, all constructed using the same NLI judge (`openai/gpt-5.2-chat`) in order to isolate variation arising from debate content rather than from the judge itself.

**Debates and motions.**

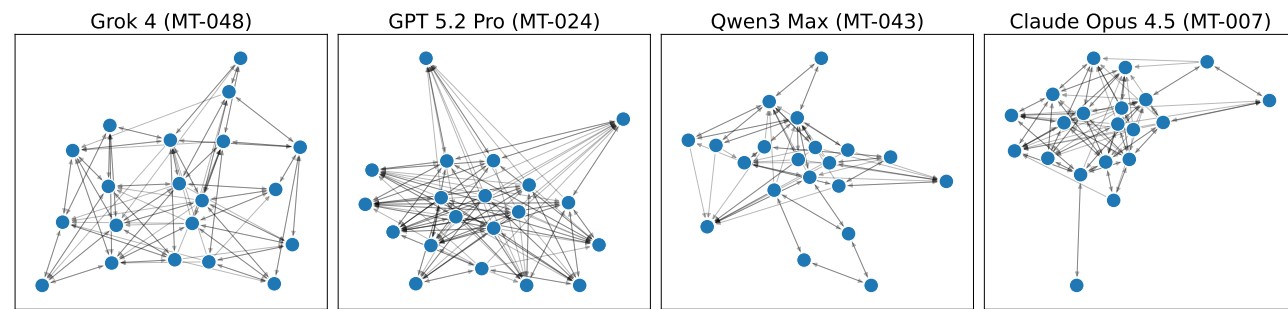

*Figure 8.* Attack graphs induced by the same NLI judge (GPT-5.2 Pro) for four multi-turn debates. Nodes correspond to arguments and directed edges indicate attacks with $W_{ij} > \tau$ (visualization threshold $\tau = 0.6$).

- **MT-048 (Grok-4):** This House would break up dominant technology monopolies.

- **MT-024 (GPT-5.2 Pro):** This House would ban the private ownership of historical artifacts.

- **MT-043 (Qwen3 Max):** This House would allow preventive detention for credible terrorism threats.

- **MT-007 (Claude Opus 4.5):** This House would abolish the minimum wage law.

**Observed densities.** Across these debates, absolute densities range from $d \approx 0.57$ to $0.65$, while mean off-diagonal attack strengths range from $\mu \approx 0.26$ to $0.37$. Thus, although all debates yield moderately dense attack graphs, the *distribution and organization* of attacks differs substantially.

**Geometric variation.** Even under a fixed judge and fixed threshold, the graphs exhibit noticeably different geometries: some debates produce tightly clustered cores with heavy reciprocal attack structure, while others display more elongated or modular layouts with peripheral argument chains. This supports the view that induced attack graphs encode meaningful structural signatures of debate content, rather than collapsing to a uniform topology.

**Interpretation.** We emphasize that these visualizations are illustrative rather than quantitative evidence of superiority of any specific topology. Their role is to demonstrate that local pairwise NLI judgments give rise to distinct global geometries, which GRASP subsequently aggregates into stable rankings. This qualitative diversity motivates our later quantitative analyses linking graph structure to convergence behavior and ranking consistency.

## K. Additional Qualitative Case Studies

We present four additional cases exhibiting the same pattern: near-unanimous high ranking under GRASP and highly dispersed, low ranking under RAW. In all the examples, #

Attackers and Mean attack strength are obtained from the $W$ constructed by the `openai/gpt-5.2-chat` judge.

**Case A (Multi-turn)**

**Motion:** This House would introduce a binding "None of the Above" option on national election ballots.

### GRASP (Structural Consensus)

**Generator:** openai/gpt-5.2-pro
**Stance:** Pro
**Angle:** Technological
**Turn:** 2
**#Attackers:** 10    **Mean attack strength:** 0.205

*A binding None of the Above can be implemented cleanly in modern election technology because it is just another selectable contest option with deterministic tabulation and a predefined trigger for reruns, reducing reliance on informal protest mechanisms like ballot spoilage that are harder to interpret and audit. It improves data quality by distinguishing abstention, undervotes, and explicit rejection, which enables better diagnostics of candidate quality and voter dissatisfaction without compromising ballot secrecy. In jurisdictions using paper-backed electronic systems, NOTA can be captured on both the human-readable ballot and the cast-vote record, supporting risk-limiting audits and minimizing disputes about intent.*

### GRASP Rankings

anthropic/claude-haiku-4.5: 1
deepseek/deepseek-v3.2: 1
google/gemini-3-flash-preview: 1
meta-llama/llama-4-scout: 3
openai/gpt-5.2-chat: 1
xiaomi/mimo-v2-flash: 3

**RAW Rankings**

anthropic/claude-haiku-4.5: 19
deepseek/deepseek-v3.2: 19
google/gemini-3-flash-preview: 8
meta-llama/llama-4-scout: 5
openai/gpt-5.2-chat: 10
xiaomi/mimo-v2-flash: 20

## Case B (Pool)

**Motion:** This House would restrict state funding only to art perceived as valuable by the general public.

**GRASP (Structural Consensus)**

**Generator:** mistralai/mistral-small-creative
**Stance:** Pro
**Angle:** Social
**#Attackers:** 29    **Mean attack strength:** 0.271

*Public funding for widely valued art maximizes the likelihood of creating accessible cultural experiences, ensuring that marginalized or economically disadvantaged groups can engage with meaningful creative expressions, thereby addressing systemic inequalities in cultural participation and enrichment.*

**GRASP Rankings**

anthropic/claude-haiku-4.5: 1
deepseek/deepseek-v3.2: 3
google/gemini-3-flash-preview: 2
meta-llama/llama-4-scout: 1
openai/gpt-5.2-chat: 2
xiaomi/mimo-v2-flash: 1

**RAW Rankings**

anthropic/claude-haiku-4.5: 11
deepseek/deepseek-v3.2: 8
openai/gpt-5.2-chat: 33
xiaomi/mimo-v2-flash: 33

## Case C (Pool)

**Motion:** This House would introduce a youth-weighted voting system in democratic elections.

**GRASP (Structural Consensus)**

**Generator:** qwen/qwen3-max
**Stance:** Pro
**Angle:** Technological
**#Attackers:** 17    **Mean attack strength:** 0.17

*Secure blockchain-based voting platforms can ensure transparency and prevent tampering in a youth-weighted system, maintaining trust while accommodating variable vote weights. These technologies already support complex voting rules in pilot programs and private governance systems.*

**GRASP Rankings**

anthropic/claude-haiku-4.5: 2
deepseek/deepseek-v3.2: 1
google/gemini-3-flash-preview: 1
meta-llama/llama-4-scout: 3
openai/gpt-5.2-chat: 1
xiaomi/mimo-v2-flash: 1

**RAW Rankings**

anthropic/claude-haiku-4.5: 20
deepseek/deepseek-v3.2: 20
google/gemini-3-flash-preview: 38
meta-llama/llama-4-scout: 17
openai/gpt-5.2-chat: 38
xiaomi/mimo-v2-flash: 39

## Case D (Pool)

**Motion:** This House would allow gene editing of embryos for non-medical traits.

**GRASP (Structural Consensus)**

**Generator:** x-ai/grok-4
**Stance:** Pro
**Angle:** Political
**#Attackers:** 21    **Mean attack strength:** 0.170

*This approach strengthens national innovation in biotechnology from a political perspective. Governments that permit such editing encourage research and development, positioning the country as a leader in global scientific progress and enhancing its geopolitical influence.*

**GRASP Rankings**

anthropic/claude-haiku-4.5: 2
deepseek/deepseek-v3.2: 1
google/gemini-3-flash-preview: 1
meta-llama/llama-4-scout: 2
openai/gpt-5.2-chat: 1
xiaomi/mimo-v2-flash: 1

**RAW Rankings**

anthropic/claude-haiku-4.5: 14
deepseek/deepseek-v3.2: 35
google/gemini-3-flash-preview: 20
openai/gpt-5.2-chat: 35
xiaomi/mimo-v2-flash: 38

**Takeaway.** Across these additional cases, we observe the same qualitative pattern: arguments that are consistently identified by GRASP as structurally central (highly ranked across all NLI models) are simultaneously relegated to low and highly dispersed ranks under RAW. These arguments tend to exhibit moderate to large numbers of attackers with non-trivial mean attack strength, indicating that GRASP promotes arguments whose importance emerges from their global position in the dialectical graph rather than from isolated surface persuasiveness. This further supports the claim that structural aggregation yields more stable and semantically grounded prioritization than direct judge rankings.

## L. DDO

A natural question is whether GRASP can serve as a proxy for persuasive success or rhetorical effectiveness. To study this, we evaluate GRASP on the Debate Decision Outcomes (DDO) dataset (Durmus & Cardie, 2018; 2019), which contains multi-round debates annotated with human votes indicating which side was more convincing, as well as point/status changes reflecting debate performance.

Our goal in this section is *not* to optimize GRASP for persuasion, but rather to test whether GRASP—designed as a structural-dialectical scoring method—implicitly correlates with convincingness.

### L.1. Dataset

We evaluate on the Debate Decision Outcomes (DDO) dataset, containing 6,967 filtered debates (Pro wins: 2,831, Con wins: 4,136). Starting from the full DDO corpus of 78,376 debates, we apply the following filters:

- Remove debates with fewer than two rounds.

- Remove debates with fewer than five human votes.

- Remove forfeited debates.

- Remove debates with unknown outcomes or fewer than three usable votes.

- Remove debates with tied convincingness labels.

| Stage | # Debates |
|---|---|
| Initial dataset | 78,376 |
| <2 rounds removed | 2,484 |
| <5 votes removed | 64,509 |
| Forfeits removed | 3,691 |
| Unknown outcome (<3 usable) removed | 189 |
| Ties removed | 536 |
| Final retained debates | 6,967 |

After filtering, the retained dataset contains 2,831 Pro-wins and 4,136 Con-wins. We then construct train/validation/test splits of 1,000 / 250 / 5,717 debates, used solely for GRASP hyperparameter selection and final evaluation.

### L.2. Experimental Setup

For each debate, we construct an argumentation graph whose weighted attack matrix $W$ is obtained using the RoBERTa-large-MNLI natural language inference model, where each entry $W_{ij}$ is the probability assigned to the *contradiction* label for the pair (argument $j$, argument $i$). The corresponding defense matrix is defined as $D = WW$, capturing two-hop counter-attack structure. GRASP scores are computed for all arguments. Side-level scores are obtained by summing argument scores per side; the higher-scoring side is predicted as the winner.

We tune GRASP hyperparameters on the validation set and select: $\alpha = 0.5, \beta = 0.25, \gamma = 1.0$. This configuration achieves validation accuracy 0.505 for convincingness and 0.514 for points/status.

We compare GRASP against the same structural approaches as defined in D.3.

We report accuracy for predicting the convincing side and the points/status winner, and Spearman correlation between GRASP score difference and convincingness margin.

### L.3. Results

We further compute the Spearman correlation between GRASP score differences ($\Delta s_{\text{GRASP}}$) and the convincingness margin provided by the dataset. We obtain $\rho = -0.009$, indicating no meaningful monotonic association between GRASP's structural preference and persuasive success.

## M. GRASP Pseudocode

*Listing 1.* GRASP fixed-point iteration (Python).

```python
import numpy as np

def safe_div(a, b, eps=1e-12):
    return a / (b + eps)

def grasp_scores(W, alpha=1.0,
     beta=0.25, gamma=0.6,
     max_iters=2000, tol=1e-10):
    #GRASP fixed-point iteration
        (unnormalized).
    W = np.maximum(W,
        0.0).astype(np.float64)
    np.fill_diagonal(W, 0.0)

    # two-step defense: paths of length
        two
    D = W @ W
    D = np.maximum(D, 0.0)
    np.fill_diagonal(D, 0.0)

    n = W.shape[0]
    s = np.ones(n, dtype=np.float64)

    for _ in range(max_iters):
        atk = W.T @ s # total incoming
            attack
        dfn = D.T @ s # total incoming
            defense (two-step)

        F = safe_div(1.0 + beta * dfn, 1.0
            + alpha * atk)
        s_next = (1.0 - gamma) * s + gamma
            * F

        if np.max(np.abs(s_next - s)) <
            tol:
            s = s_next
            break
        s = s_next

    return s

def grasp_ranking(W, **kwargs):
    s = grasp_scores(W, **kwargs)
    order = np.argsort(-s) # best first
    return order, s
```

| Arg | Stance | Angle | #Attackers | Mean atk. $\mu$ | Text |
|---|---|---|---|---|---|
| arg_19 | Pro | Social | 10 | 0.6850 | Dominant technology monopolies exacerbate social inequalities by prioritizing content that favors affluent users and marginalizing voices from lower socioeconomic groups. Breaking them up would enable smaller platforms to cater specifically to diverse demographics, fostering more inclusive online communities and reducing digital divides. This change would also improve social cohesion by decentralizing control over algorithms that currently amplify polarizing content. |
| arg_2 | Con | Technological | 12 | 0.6400 | Dominant technology monopolies drive technological innovation by concentrating resources for large-scale research and development that smaller entities cannot match, countering the claim that they suppress advancements. Breaking them up would fragment essential platforms and standards, potentially slowing the integration of new technologies across ecosystems. This fragmentation could limit rather than promote broader access to cutting-edge solutions, as coordinated efforts by monopolies often accelerate widespread adoption. |
| arg_4 | Con | Political | 12 | 0.5683 | Dominant technology monopolies provide a centralized point for political accountability, allowing governments to engage with fewer entities for effective oversight rather than dealing with fragmented influences from multiple smaller companies. Breaking them up would likely increase overall lobbying efforts as numerous firms compete for policy favors, potentially overwhelming democratic processes instead of diluting corporate power. This concentration enables more efficient implementation of regulations that address public welfare without the chaos of dispersed political pressures. |
| arg_15 | Pro | Economic | 10 | 0.6700 | Dominant technology monopolies distort economic efficiency by capturing excessive profits through barriers to entry that prevent efficient resource allocation across industries. Breaking them up would allow for more dynamic markets where new entrants can compete on merit, leading to improved productivity and broader economic distribution of wealth. This change would also reduce the risk of market failures associated with over-reliance on a few firms for critical technological services. |
| arg_17 | Pro | Moral | 10 | 0.6970 | Dominant technology monopolies create moral issues by enabling the exploitation of user data without sufficient accountability, which undermines trust in digital systems and harms individual dignity. Breaking them up would foster a more ethical environment where multiple companies must compete on the basis of responsible practices rather than relying on unchecked dominance. This division would also reduce the moral risks associated with concentrated control over information flows that can amplify societal divisions. |
| arg_11 | Pro | Social | 10 | 0.5990 | Dominant technology monopolies contribute to social isolation by designing platforms that prioritize addictive engagement over meaningful interactions among users. Breaking them up would encourage the development of diverse social networks that facilitate healthier community building and reduce echo chambers. This restructuring would also enhance social equity by allowing smaller entities to address the needs of underserved populations more effectively. |

*Table 9.* Metadata and full text for the most rank-volatile arguments in debate MT-048 (rank dynamics shown in Fig. 4). # Attackers and Mean attack strength $\mu$ are obtained via the $W$ constructed by the `openai/gpt-5.2-chat` judge model. $\mu$ is computed over incoming attackers with $W_{ij} > 0$.

| Method | Conv. Acc. | Pts./Status Acc. |
|---|---|---|
| GRASP | 0.493 | 0.494 |
| GRASP-W$_\infty$ | 0.500 | 0.502 |
| GRASP-W$_1$ | 0.499 | 0.503 |
| GRASP-W$_\infty$+$\bar{D}$ | 0.495 | 0.497 |
| GRASP-W$_1$+$\bar{D}$ | 0.491 | 0.493 |
| H-Categorizer | 0.508 | 0.510 |
| Binary In-Degree | 0.507 | 0.510 |
| Max In-Degree | 0.508 | 0.510 |
| Katz Centrality | 0.508 | 0.512 |

*Table 10.* Performance on DDO. All methods operate near chance level. GRASP variants achieve accuracy comparable to simple structural baselines but do not outperform them.

