# OpenReview forum: "GRASP: Deterministic Argument Ranking in Natural-Language Debates via Attack–Defense Propagation"
_ICML.cc/2026/Conference — Submitted to ICML 2026_

### Official Review · Reviewer_Ujxp · 2026-02-23

**Soundness:** 4
**Presentation:** 4
**Significance:** 3
**Originality:** 3
**Overall Recommendation:** 5
**Confidence:** 4

**Summary:**

This paper proposes a framework based off Dung-style abstract argumentation frameworks to improve LLMs as holistic debate judges. Proofs of convergence and empirical results are provided.

**Compliance With Llm Reviewing Policy:**

Affirmed.

**Key Questions For Authors:**

How does this approach relate to other models of argumentation, such as argumentation games?
To put it in another way, how applicable would this be when the set of arguments $A$ is unknown/unseen?

**Limitations:**

Yes

**Strengths And Weaknesses:**

This is an excellent paper. It is very well-written, motivated, and technically somewhat strong. I only have a few minor points of feedback:

1. In terms of methodology/reproducibility, it is customary to use the model names, and not the links (?).
    - This also means you need to add citations to the model's original works; and disclose versions, as well as parameter calls like temperature and maximum tokens.
    - This is also related to L325, C2 (what is `mt_048_x-ai__grok-4`?)
2. For Theorem 3.1, a proof sketch would be quite helpful.
3. This is not the first work to determine/find low agreement in LLMs-as-judges, especially in debates. I recommend removing this contribution from this section--and perhaps adding these works to the related works section. The other three contributions of the paper are quite strong already in my opinion.
    - This is also particularly important because what this paper finds is that a _prompt_ has low IAA. To make a full claim on LLMs-as-judges having low agreement, you should also run some form of automated prompt optimisation / try other methods.

There are two key limitations of this work that are not sufficiently discussed, in my opinion:
- The data generation process is quite homogenous, with the models performing self-play. This makes the dataset future useability somewhat limited.
- Dung's framework / AAFs have some shortcomings, particularly around realism. This needs to be further discussed.

---

> ### Author Rebuttal · Authors · 2026-03-30
>
> We sincerely thank the reviewer for the very positive and thoughtful review. We are especially grateful that the reviewer found the paper well-written, well-motivated, and technically strong.
>
> ### Clarity on reproducibility / methodology
> We thank the reviewer for pointing this out. In revision, we will make internal identifiers explicit; for example, `mt_048_x-ai__grok-4` denotes the 48th multi-turn debate, generated by Grok 4. We will also cite all models appropriately, report model settings, and improve dataset documentation. We also plan to open-source the dataset.
>
> ### Proof Sketch of Theorem 3.1
>
> The strategy is to identify a bounded, invariant subset $S$ and show that $G$ is a contraction on $S$, with respect to the infinity norm. Under those conditions, the Banach fixed-point theorem yields existence and uniqueness of the solution found by GRASP.
>
> First, we define $S = \\{\mathbf{s} \in \mathbb{R}^d : \\|\mathbf{s} - \mathbf{1}\\|_\infty \leq 1\\}$.
>
> Under the assumption that the entries of $W,D$ are non-negative, we show in the appendix that $\alpha \leq \frac{1}{4\\|W\\|_1}$ and $\beta \leq \frac{1}{4\\|D\\|_1}$.
>
> Together with the definition of $S$, these conditions imply $\\|G(\mathbf{s}) - \mathbf{1}\\|_\infty \leq 1$.
>
> Hence if $s \in S$, then $G(s) \in S$ (invariance).
>
> Next, we study the contraction properties of $G$, i.e. how close vectors $x,y \in S$ get after $G$ is applied.
>
> The ratio structure of $G$ prevents direct spectral arguments, yet we show that upper-bounding $\\|G(x)-G(y)\\|_\infty$ becomes possible after some algebraic manipulations.
>
> Under the same assumption set that guarantees invariance, we can conclude that
> $\\|G(x)-G(y)\\|_\infty \le \frac{3}{4}\\|x-y\\|\_\infty$.
>
> Hence $G$ is a contraction, and we can conclude by applying the Banach fixed-point theorem.
>
> ### Clarifying the contribution claim on LLM-as-judge disagreement
> We thank the reviewer for this important point. We will revise the contribution statement accordingly. More precisely, the empirical result in our paper is that low inter-model agreement among LLM judges persists under the natural-language debate setup we study. We use this observation primarily as motivation for shifting from holistic judging to local pairwise interaction judgments followed by structure-aware aggregation. We will therefore soften the framing and more clearly acknowledge prior work on LLM-as-judge disagreement. We also appreciate the reviewer’s observation that the other three primary contributions are strong independently of this framing.
>
> ### On the homogeneity of the benchmark
> The current generation pipeline is indeed relatively homogeneous, since it relies on controlled model-generated debates. This was intentional: STRUCTDEBATE was designed as a controlled testbed to isolate structural effects and enable reproducible comparison across models and settings. GRASP itself, however, is agnostic to the source corpus: given an interaction graph, it defines a structure-aware ranking whether the underlying arguments are synthetic or real-world. We will make this distinction clearer and emphasize that the homogeneity is a property of the benchmark design rather than of the operator.
>
> ### Limitations of Dung's framework
> We thank the reviewer for this thoughtful point. Dung-style argumentation serves as the motivating foundation for GRASP, while our contribution is to adapt that structural view to natural-language debates via weighted interaction graphs induced from NL arguments. The paper is therefore not claiming that classical AAFs alone are a fully realistic model of debate; rather, it uses their attack/defense intuition as the basis for a method intended for richer natural-language settings. We will clarify this distinction more explicitly in revision.
>
> ### Relation to argumentation games and unseen arguments
> We thank the reviewer for the insightful question. GRASP is currently intended as a ranking semantics on an instantiated interaction graph, consistent with the paper’s notion of structural sufficiency, where robustness is evaluated with respect to the arguments explicitly present in the graph. Argumentation-game style models are especially relevant in settings where arguments are revealed dynamically or remain initially unseen. We view this not as outside the scope of GRASP, but as a promising extension direction: a dynamic version of GRASP could update rankings as new arguments and interactions arrive over time. The current paper focuses on the static setting, while such evolving debate environments are a natural avenue for future work.
>
> We sincerely thank the reviewer again for the generous and constructive feedback. We are grateful for the strong support of the paper’s main contributions, and hope the clarifications further sharpen the framing, scope, and presentation of the work.

---

> > ### Author Rebuttal · Reviewer_Ujxp · 2026-04-03
> >
> > I appreciate the author's responses and consider my concerns fully resolved.

---

> > > ### Author Response · Authors · 2026-04-05
> > >
> > > Thank you very much for the follow-up and for confirming that the concerns have been fully resolved. We are very grateful for your positive assessment of the paper and its core contributions. Your support is sincerely appreciated.

---

### Official Review · Reviewer_1D2c · 2026-03-11

**Soundness:** 2
**Presentation:** 2
**Significance:** 2
**Originality:** 3
**Overall Recommendation:** 2
**Confidence:** 2

**Summary:**

This paper proposes a method that can rank arguments with respect to its "strength". They define the structural sufficiency concept, leading to define the strength of an argument depending on the counter arguments of its counter arguments. They evaluate their algorithm on an artificial dataset they generated, looking at the discrepancy of the results between 6 LLMs. They compare to an argument strength model using the raw text of the debate in its prompt.

**Compliance With Llm Reviewing Policy:**

Affirmed.

**Key Questions For Authors:**

* What does motivate the 6 semantic angles?

**Limitations:**

yes

**Strengths And Weaknesses:**

Strengths:

* Authors define the concept of structural sufficiency
* Generation of a dataset of argument and counter arguments on various topics
* Proposed algorithm shows more similar outputs in between the 6 LLMs
* This work does look original.

Weaknesses:
* The paper lacks grounding on arugmentation mining theory.
* The argument's strength is central in this work but not define well.
* The method shows LLM are consistent, but not that they are correlated with manual annotation somehow, which questions the usability of the method.
* It seems the authors do not compare with other attack-based ranking methods.
* Missing litterature, in particular from Benno Stein's group and the Touché's tasks that have been working on argument ranking.
* The Arguments from the dataset they use are generated. It seems like they can contain more than one unique argument (see Appendix K for example) * No human validation of the edges of the graph
* Dataset would benefit from an in-depth analysis of its content
* The paper is pretty hard to follow in its structure.
* The disparity in prediction of argument strength in between the LLMs is interesting. However it would be more robust to see a proposal grounded in argumentation and linguistic theories, with manual annotations/validations.

---

> ### Author Rebuttal · Authors · 2026-03-30
>
> We thank the reviewer for the detailed feedback and for recognizing the paper’s originality and broad empirical evaluation. Below we clarify the intended scope of the paper, address points that appear to stem from misunderstandings of the submission, and indicate how we will improve the presentation.
>
> ### Arg. mining vs. GRASP
> GRASP is not a contribution to argument mining, - “the automatic identification and extraction of the structure of inference and reasoning expressed as arguments presented in natural language” [1]. Arg. mining traditionally focuses on identifying argumentative units (claims, evidences) and their relations within a text. By contrast, GRASP addresses a downstream problem: once debate arguments have been instantiated as an interaction graph, how should they be ranked? $W$ construction may involve arg-mining-style components, but GRASP is complementary to that stage and operates on the resulting cross-argument graph.
>
> ### Arg. strength / DDO labels
> Our use of “argument strength” is distinct from persuasive quality, truth, or human convincingness. Our intended notion is **structural robustness** with respect to the instantiated interaction graph: an argument is stronger insofar as it faces less incoming attack and is better defended within that graph. This is formalized in Sec. 3 through structural sufficiency, which GRASP captures in graded form. This connection is made in the paper through cited work from **Benno Stein’s group** (**Wachsmuth et al. (2017)**), and we will make it more explicit.
>
> Likewise, lack of correlation with the DDO manual labels is not evidence against usability, because they measure convincingness or persuasive success rather than the target quantity GRASP is designed for. As stated in the paper, the DDO analysis was included precisely to test this distinction.
>
> ### Attack-based baselines / cited literature
> We already compare against multiple attack-based baselines in Appendix D, including H-Categorizer, KatzAttack, Defense Ratio, Binary Indegree, and Max Incoming Attack, all on the same attack matrix $W$. H-Categorizer is a ranking semantics from abstract argumentation, which we adapt to our natural-language setting as a direct structural baseline. We show GRASP attains the lowest structural-sufficiency violation rate. If the reviewer has a specific additional operator in mind for the natural-language setting, we would be happy to include it.
>
> The paper also already cites multiple works from **Benno Stein’s group**, including **Wachsmuth et al. (2017)**, **Mirzakhmedova et al. (2023)**, and **Mirzakhmedova et al. (2024)**. We are happy to add further specific references if the reviewer has particular works in mind.
>
> ### STRUCTDEBATE / edge validation
> **STRUCTDEBATE** is intentionally defined over full debate turns rather than individual argumentation units. Thus, it containing multiple unique arguments is not a flaw of its design; it reflects how arguments are naturally presented in standard debates. We suspect part of the confusion here may stem from reading the paper as an arg-mining setup, where finer-grained decomposition is central. Our setting is different: GRASP operates over cross-argument interactions in debates rather than the internal decomposition of a single argument.
>
> Human validation of graph edges would be valuable. Such annotation would offer an alternative way to instantiate the interaction graph, while GRASP itself remains unchanged once the graph is given.
>
> ### Dataset / presentation / theory
> We will add more detailed analysis of STRUCTDEBATE and plan to open-source the dataset.
>
> Reviewer 3P7H and Reviewer Ujxp both found the paper well-written and easy to follow. We nevertheless appreciate the reviewer raising this point, and would be grateful for any concrete suggestions to further strengthen the structure and presentation.
>
> The proposal is already grounded in argumentation theory, namely abstract argumentation and ranking semantics. GRASP is motivated by attack/defense structure and the structural sufficiency principles formalized in the paper, and extends this perspective to natural-language debates through weighted interaction graphs and a convergent defense-aware operator. If the reviewer has a specific linguistic framework or area in mind, we would be grateful for the suggestion.
>
> ### Motivation for 6 angles
> We arrived at the six angles—economic, legal, moral, political, social, and technological—through a qualitative review of related work and available datasets, followed by discussion of recurring themes in debate topics.
>
> We thank the reviewer again for the detailed feedback. We hope these clarifications address the main concerns and make the strengths of the work clearer, including several aspects also viewed positively by Reviewers 3P7H and Ujxp. We hope this will lead to a more favourable assessment of our work.
>
> [1] Lawrence, John, and Chris Reed. “Argument mining: A survey.” *Computational Linguistics* 45, no. 4 (2019): 765–818.

---

> > ### Author Rebuttal · Reviewer_1D2c · 2026-04-02
> >
> > I maintain my score as the weaknesses that have been pointed out are major and have not been resolved. It lacks of grounding in argument mining field, or adaptation of the method to other field if it is general. Finally, the data is fully artificial and there is no human validation.
> >
> > However the idea is interesting and would require another round of revision to reach a satisfying level.
> >
> > If the review can seem harsh and missing some references (I am out of office, but willing to finish the review process) I can testify that imho this work is not at the level for an ACL conference because of several lacks. It might be enough for ICML but I doubt it. The meta reviewer will judge the paper's quality.
> >
> > Some answers below:
> >
> > # Arg. mining vs. GRASP
> > The paper principally apply the method to argument mining. If the method is broader than this field, several experiment on other dataset should be executed. If not, the authors should present precise their work in the argument mining state of the art.
> >
> > # Arg. strength / DDO labels
> > The assummption that all the possible arguments are inside the corpus/graph is very strong and not necessarily true.
> >
> > # Motivation for 6 angles
> > This does not seem motivated enough at all in the paper, and no more information is available in the rebuttal.

---

> > > ### Author Response · Authors · 2026-04-05
> > >
> > > Thank you for the follow-up. We believe the remaining points mainly reflect a few continuing misunderstandings of the paper’s scope and setup, which we clarify below.
> > >
> > > ### Arg Mining vs GRASP (arg ranking)
> > >
> > > We respectfully disagree. As clarified in our main rebuttal, **GRASP is not a contribution to argument mining, but to argumentation ranking semantics**. Argument mining is commonly defined as identifying argumentative units and their relations from natural-language text or discourse [1,2]. GRASP addresses a different problem: **once debate arguments have been instantiated as an interaction graph, how should they be ranked?** It operates on that instantiated **cross-argument interaction graph** and defines a convergent defense-aware ranking operator over it. This distinction is standard in the literature on argument mining versus ranking-based semantics [1,2,3,4,5,6], and we will make it even clearer in the revision.
> > >
> > > [1] Lawrence, John, and Chris Reed. “Argument mining: A survey.” *Computational Linguistics* 45, no. 4 (2019): 765–818.
> > >
> > > [2] Peldszus, Andreas, and Manfred Stede. "From argument diagrams to argumentation mining in texts: A survey." International Journal of Cognitive Informatics and Natural Intelligence (IJCINI) 7, no. 1 (2013): 1-31.
> > >
> > > [3] Bonzon, Elise, Jérôme Delobelle, Sébastien Konieczny, and Nicolas Maudet. "A comparative study of ranking-based semantics for abstract argumentation." In Proceedings of the AAAI Conference on Artificial Intelligence, vol. 30, no. 1. 2016.
> > >
> > > [4] Heyninck, Jesse, Badran Raddaoui, and Christian Straßer. "Ranking-based Argumentation Semantics Applied to Logical Argumentation." In IJCAI, pp. 3268-3276. 2023.
> > >
> > > [5] Amgoud, Leila, Jonathan Ben-Naim, Dragan Doder, and Srdjan Vesic. "Ranking Arguments With Compensation-Based Semantics." KR 16 (2016): 12-21.
> > >
> > > [6] Mailly, Jean-Guy, and Julien Rossit. "Argument, I choose you! preferences and ranking semantics in abstract argumentation." In Proceedings of the International Conference on Principles of Knowledge Representation and Reasoning, vol. 17, no. 1, pp. 647-651. 2020.
> > >
> > > ### On the instantiated graph assumption
> > >
> > > The paper does **not** assume that “all possible arguments” are present in the graph in any universal sense. Rather, the setup is explicit and debate-specific by design: if a debate contains ($n$) arguments, the corresponding interaction graph contains ($n$) nodes and ($n^2$) possible directed relations among them. GRASP evaluates **structural robustness relative to that instantiated graph**. Thus, the paper’s notion of structural sufficiency concerns the observed debate graph, not an exhaustive representation of every conceivable argument outside it. This is precisely the distinction the paper draws between global sufficiency and our proposed structural sufficiency (Section 4, L119-127).
> > >
> > > ### Motivation for the 6 angles
> > > As clarified in our main rebuttal, the six angles were introduced as **controlled prompting dimensions** to induce diverse but comparable argument pools for each motion/side, rather than near-paraphrases. Their role is methodological: to broaden the interaction graph while keeping generation structured and reproducible. The categories were chosen through a **qualitative design process** based on reviewing related datasets and debate materials and identifying recurring thematic frames in debate topics. This is a standard qualitative design approach in social computing (primary field of our submission), HCI, psychology, and related applied fields [7,8,9].
> > > They were intended as a compact design choice for coverage and control, not as an exhaustive ontology.
> > >
> > > [7]McDonald, Nora, Sarita Schoenebeck, and Andrea Forte. "Reliability and inter-rater reliability in qualitative research: Norms and guidelines for CSCW and HCI practice." Proceedings of the ACM on human-computer interaction 3, no. CSCW (2019): 1-23.
> > >
> > > [8] Braun, Virginia, and Victoria Clarke. "Using thematic analysis in psychology." Qualitative research in psychology 3, no. 2 (2006): 77-101.
> > >
> > > [9] Glaser, Barney, and Anselm Strauss. Discovery of grounded theory: Strategies for qualitative research. Routledge, 2017.
> > >
> > > ### On human validation and the operator contribution
> > > Human validation of graph edges would certainly be a valuable complementary direction. At the same time, the primary contribution of the paper is the **GRASP operator itself**, which is defined on an instantiated interaction graph and is **agnostic** to the underlying data source or graph-construction route. That is the **primary methodological contribution** of the submission.
> > >
> > > We would also be grateful for any concrete references, baselines, or actionable suggestions the reviewer believes would better align the paper with the remaining concerns.
> > >
> > > We hope this addresses the remaining concerns and helps the reviewer reassess the paper in light of these clarifications, the strengths already recognized in this review, and the positive assessments from Reviewers 3P7H and Ujxp.

---

### Official Review · Reviewer_3P7H · 2026-03-11

**Soundness:** 4
**Presentation:** 3
**Significance:** 3
**Originality:** 4
**Overall Recommendation:** 5
**Confidence:** 4

**Summary:**

The authors introduce the GRASP framework, a method for ranking arguments strictly based on their "attack" and "support" relations. The GRASP framework involves using judge LLMs to assess attacks between arguments and iteratively updating a damped vector to extract higher-order relations post-propagation (the authors provide a proof showing that this vector converges to a stable ranking). This follows the literature in that an argument is strong if it negates its attackers. The authors propose the notion of "structural sufficiency", explaining that the arguments should only be assessed on their explicit relations, not anticipated counter-arguments, appeal, or persuasion. The experimental setup involves over 7000 arguments, providing strong evidence that GRASP increases inter-model agreement across debates. The authors show that the attack matrices are highly correlated across models, indicating the strength of their method. Finally, two case studies are presented, providing further proof of the validity of structural sufficiency.

**Compliance With Llm Reviewing Policy:**

Affirmed.

**Final Justification:**

The authors' addressed my main concerns in the rebuttal. It is clear that this paper is thorough, and the questions I had regarding some methodological choices (i.e. $D = W^2$), the case studies, and the purpose of GRASP were answered comprehensively.

As stated before, I believe this paper is well-written and clear. The authors present several theorems and definitions (e.g. structural sufficiency, GRASP update rule) that are easy to follow. My only criticism on the presentation is that the paper could be better positioned, which was addressed in the rebuttal. The actual content of the paper is novel and original. The central question of using local pairwise arguments to produce final rankings is interesting, and the authors produced an innovative methodology to answer this question. In addition, GRASP is well-justified and supported by the proofs and studies in the appendices. Further, the discussion on the results is insightful and supports the presented method. I also believe that this contribution is of significance as there is importance in evaluating the consistency of LLM arguments.

Given the paper and the rebuttal, my final score is a 5/6.

**Key Questions For Authors:**

See questions in the weakness section.

**Limitations:**

yes

**Strengths And Weaknesses:**

Strengths:
- The paper is comprehensive, providing a clear methodology and controlling for various factors such as GRASP variants, judge LLMs, and agreement metrics. I also believe that it is well-written, well-detailed, and relatively easy to follow.
- The GRASP operator is highly novel, original, and creative. I think it is a meaningful contribution to the area of computational argumentation. Evaluation based on local, semantic interactions provides a much stronger means of evaluating arguments rather than holistic judging. The authors harnessed this fact and developed a method that elegantly converges to an accurate ranking of arguments.
- The authors provide strong empirical results of the GRASP framework as compared to the RAW baseline. The large dataset and various controls/variants (generators, debate settings, etc.) validate these results.
- The discussion and analysis of the results is insightful, explaining the behavior of GRASP and its emphasis on structural sufficiency.
- The paper addressed an important issue-- LLMs are increasingly used as judges and we need to ensure the consistency of their assessments. The authors proposed a framework that met this need, and the significance of this cannot be understated.

Weaknesses:
- One of the main issues I see is the high Pearson correlation across model attack matrices. If the attack matrices are similar, then would it not be possible for the success of GRASP to be mainly a factor of the pairwise relations between arguments? I think the authors addressed this, but they should have compared GRASP to a method that only involves creating attack matrices and rankings based on those said matrices. An extension of this is adding human judges to compare performance to LLMs.
- I also think that the paper could be better situated against the literature. There should be more discussion regarding the background of assessing arguments on persuasion, imagined counter-arguments, and particularly attack/support structure.
- The initialization of D = WW might be problematic, and it limits the potential of the GRASP method capturing multi-hop and higher order relations. The authors addressed this in the limitations, but I would have liked to see some more justification regarding this choice and perhaps a comparison to alternative ways to initialize D.
- In the real world, arguments are more complex than those in the dataset. How well will GRASP generalize "in the wild"? I am referring to the fact that it might be more subjective and difficult to assess pairwise arguments that can be both contradictory and complex.
- Lastly, the case studies are a bit unclear. I am slightly confused on the notion that attack matrices are correlated across models, but attack graphs for the same debate are model-dependent (section 5.6). Is this not somewhat contradictory? I understand that 5.6 shows GRASP's dependency on structure, but I would appreciate some clarification of how sections 5.5 and 5.6 relate to each other.

---

> ### Author Rebuttal · Authors · 2026-03-30
>
> We sincerely thank the reviewer for the thoughtful and encouraging review. We especially appreciate the recognition that GRASP is novel, that local semantic interaction judgments provide a stronger basis than holistic judging, and that the empirical study is broad. We also thank the reviewer for noting the paper’s clarity and the discussion of structural sufficiency.
>
> ### High Pearson similarity of $W$ across models and alternate ranking operators.
> A central perspective of the paper is that moving from global/holistic evaluation to local pairwise interaction judgments is itself valuable: judges may disagree substantially on end-to-end rankings while agreeing much more on whether one argument attacks another. This is consistent with Table 2, Sec. 5.7, and Appendix K: global rankings are weakly correlated, while local pairwise judgments are more stable. The high Pearson correlation across attack matrices therefore indicates that this local substrate is relatively stable. Appendix E.2 further shows that more similar induced attack graphs lead to more similar GRASP rankings.
> GRASP then addresses a separate question: *given a fixed graph $W$, how should these local judgments be converted into a final ranking?* It combines direct incoming attack with defense-aware structural information from the graph, rather than relying only on raw pairwise attack counts. Accordingly, Appendix D compares GRASP to alternative same-$W$ ranking operators and shows that GRASP attains the lowest structural-sufficiency violation rate.
> We also agree that incorporating human judges would be valuable. This would test the same hypothesis in a broader setting and provide an alternative graph-construction route. We aim to explore this as a future direction to extend our work.
>
> ### Positioning of the paper.
> GRASP primarily contributes to argumentation ranking semantics: given an interaction graph, it defines a convergent defense-aware ranking operator. We will distinguish this more clearly from rhetorical and dialectical evaluation notions such as convincingness and persuasiveness, since GRASP targets structural robustness rather than persuasive effectiveness.
>
> ### On the choice $D=W^2$.
> We use $D=W^2$ because, in an attack graph, a two-step path naturally encodes defense: one argument attacks an attacker of another. Because GRASP computes scores recursively, higher-order defense chains can still influence the fixed point. More generally, GRASP is not tied to $D=W^2$: the operator and convergence result apply to any nonnegative bounded $W$ and $D$ satisfying the stated assumptions. We use $W^2$ here because it is the most natural choice for debate-style attack graphs, while alternative constructions and domain-specific choices of $D$ are natural future directions.
>
> We also evaluated $D=W^T$, $D=W^4$, and $D=W^2+\frac{1}{2}W^4$ on the Structural Graphs evaluation in Appendix D. The results below show that $D=W^2$ still gives the lowest violation rate and mean severity. For higher-order variants, $W$ and $D$ were normalized.
>
> | Method | Violation | Severity | Iterations | Converged |
> | -------- | -------- | -------- | -------- | -------- |
> | GRASP ($D=W^2$) | 0.003 | 0.010 | 65.8 | Yes |
> | GRASP ($D=W^T$) | 0.163 | 0.091 | 57.6 | Yes |
> | GRASP ($D=W^4$) | 0.220 | 0.117 | 60.6 | Yes |
> | GRASP ($D=W^2+\frac{1}{2}W^4$) | 0.269 | 0.129 | 61.2 | Yes |
>
> ### GRASP in the wild.
> We agree that real-world arguments can be more complex than those in our benchmark. STRUCTDEBATE was intentionally designed as a controlled testbed: although machine-generated, it is grounded in 50 real-world motions from DebateData.io and uses controlled prompts/settings to isolate structural effects and enable reproducible comparison. GRASP itself is agnostic to the corpus: given $W$ and $D$, it defines an interpretable structure-aware ranking operator. Our experiments on STRUCTDEBATE already demonstrate the main empirical benefits of the method in a natural-language debate setting. Obtaining reliable interaction graphs in the wild is non-trivial, but this challenge lies primarily in graph construction rather than in the operator itself.
>
> ### Clarification on Secs. 5.5 and 5.6.
> Section 5.5 studies agreement in the overall pattern of pairwise attack scores, while Section 5.6 studies which edges remain after thresholding. Here, “high-confidence” means scores above the threshold $\tau$ used to build the graph. As discussed in L353–366, judges may agree strongly on the former while differing substantially in thresholded density and mean positive attack strength. High Pearson similarity of $W$ therefore does not imply identical high-confidence graph structure.
>
> We thank the reviewer again for the thoughtful and constructive feedback. We hope these clarifications make the paper’s framing and contributions clearer, and we appreciate the opportunity to strengthen the presentation accordingly.

---

> > ### Author Rebuttal · Reviewer_3P7H · 2026-04-03
> >
> > I appreciate the authors' helpful responses to my questions and points.
> >
> > I understand that GRASP addresses the question: "given a fixed graph $W$, how should these local judgements be converted into a final ranking?". The graph $W$ is dependent on local pairwise attack counts (and notably so), but the question is how to convert those counts into rankings. The authors' explanation of this further solidifies the contribution and novelty of GRASP.
> >
> > The choice of $D = W^2$ is justified given the recursive nature of GRASP.
> >
> > I thank the authors for explaining sections 5.5 and 5.6.
> >
> > I am confident in GRASP's technical soundness and believe the paper is thorough, diligent, and novel. Thus, I will change my score to a 5.

---

> > > ### Author Response · Authors · 2026-04-05
> > >
> > > Thank you very much for the thoughtful follow-up and for engaging so carefully with our rebuttal. We are especially grateful that our clarifications helped make the contribution and novelty of GRASP clearer, particularly the distinction between local pairwise judgments and the ranking operator itself, the justification for $D=W^2$, and the discussion of Sections 5.5 and 5.6. We also sincerely appreciate your confidence in the paper’s technical soundness and your decision to raise the score.

---

### Decision · Program_Chairs · 2026-04-30

**Decision:**

Reject

**Comment:**

**Summary**
The LLM judge of debates, if done with one score only, are unstable. This paper instead uses a framework, GRASP, that aggregates attack-defense graphs into a global ranking. The local interactions are more reproducible across models, and do not correlate the unstable human "convincingness" labels.

**Strengths**
- The problem is motivated well.
- The proposed GRASP framework has strong stability across models.
- The case study that contrasts to human convincingness scores strengthens the contribution.

**Weaknesses**
- There are some concerns regarding the choice of the framework's technical details, e.g., D=WW, strength of the arguments, proof sketch of theorems. In the rebuttal, many of them are addressed. It'd be great to incorporate these edits into the paper.
- Similarly, the case study can be clarified more.
- The benchmark used in this paper appears a bit homogeneous, and more in-depth analysis of the content, (or a human validation) would strengthen this paper.
- The paper can be better grounded in the literature on argument theory, argumentation ranking semantics, debate analysis, and realism. It'd be great to also discuss the strengths/shortcomings against the literature.

Overall, this is a paper with solid contributions, but there are several weaknesses. It'd be great if a set of reviewers go through the new version of the paper once these edits are incorporated, in the next venue.